# Constructing a measurement-based spatially explicit inventory of US oil and gas methane emissions (2021)

**Mark Omara**[1,2], **Anthony Himmelberger**[2], **Katlyn MacKay**[1], **James P. Williams**[1], **Joshua Benmergui**[1,2,3], **Maryann Sargent**[3], **Steven C. Wofsy**[3], **and Ritesh Gautam**[1,2]

[1]Environmental Defense Fund, New York, NY 10010, USA
[2]MethaneSAT, LLC, Austin, TX 78701, USA
[3]School of Engineering and Applied Sciences CE1, Department of Earth and Planetary Science, Harvard University, Cambridge, MA 02138, USA

**Correspondence:** Mark Omara (momara@edf.org) and Ritesh Gautam (rgautam@edf.org)

**Abstract.** Accurate and comprehensive quantification of oil and gas methane emissions is pivotal in informing effective methane mitigation policies while also supporting the assessment and tracking of progress towards emissions reduction targets set by governments and industry. While national bottom-up source-level inventories are useful for understanding the sources of methane emissions, they are often unrepresentative across spatial scales, and their reliance on generic emission factors produces underestimations when compared with measurement-based inventories. Here, we compile and analyze previously reported ground-based facility-level methane emissions measurements ($n = 1540$) in the major US oil- and gas-producing basins and develop representative methane emission profiles for key facility categories in the US oil and gas supply chain, including well sites, natural-gas compressor stations, processing plants, crude-oil refineries, and pipelines. We then integrate these emissions data with comprehensive spatial data on national oil and gas activity to estimate each facility's mean total methane emissions and uncertainties for the year 2021, from which we develop a mean estimate of annual national methane emissions resolved at $0.1° × 0.1°$ spatial scales ($\sim 10\,\text{km} × 10\,\text{km}$). From this measurement-based methane emissions inventory (EI-ME), we estimate total US national oil and gas methane emissions of approximately 16 Tg (95 % confidence interval of 14–18 Tg) in 2021, which is $\sim 2$ times greater than the EPA Greenhouse Gas Inventory. Our estimate represents a mean gas-production-normalized methane loss rate of 2.6 %, consistent with recent satellite-based estimates. We find significant variability in both the magnitude and spatial distribution of basin-level methane emissions, ranging from production-normalized methane loss rates of $< 1\,\%$ in the gas-dominant Appalachian and Haynesville regions to $> 3\,\%$–6 % in oil-dominant basins, including the Permian, Bakken, and the Uinta. Additionally, we present and compare novel comprehensive wide-area airborne remote-sensing data and results for total area methane emissions and the relative contributions of diffuse and concentrated methane point sources as quantified using MethaneAIR in 2021. The MethaneAIR assessment showed reasonable agreement with independent regional methane quantification results in sub-regions of the Permian and Uinta basins and indicated that diffuse area sources accounted for the majority of the total oil and gas emissions in these two regions. Our assessment offers key insights into plausible underlying drivers of basin-to-basin variabilities in oil and gas methane emissions, emphasizing the importance of integrating measurement-based data when developing high-resolution spatially explicit methane inventories in support of accurate methane assessment, attribution, and mitigation. The high-resolution spatially explicit EI-ME inventory is publicly available at https://doi.org/10.5281/zenodo.10734299 (Omara, 2024).

# 1 Introduction

Accurate characterization of oil and gas methane emissions across spatial scales – from the facility level to the basin and national level – is an essential component of methane reduction programs that are integral to mitigating the near-term catastrophic impacts of human-induced global warming (IPCC, 2021). As governments, industry, and various stakeholders publicly commit to cut their methane emissions footprints (OGCI, 2021; GMP, 2021), accurate methane inventories will play a crucial role in the development and implementation of effective methane reduction approaches as well as in tracking progress toward emission reduction targets.

At the national level, methane inventories are typically developed using "bottom-up" methods; for example, these methods are used by most countries that report annual greenhouse gas inventories to the UNFCCC (UNFCCC, 2023). Bottom-up methane inventories are developed by applying generic (or, in some cases, empirically determined) component- or source-level emission factors to national oil and gas activity data (EPA, 2022). While these inventories are useful as first-order estimates of the emission sources, they often lack the accuracy needed to characterize methane emissions, their sources, and their trends over time at the facility scale to the basin level.

In addition, scores of recent studies that have focused on specific oil and gas basins (Zhang et al., 2020), specific countries (Alvarez et al., 2018; Shen et al., 2021; Zavala-Araiza et al., 2021; Johnson et al., 2023), and the analysis at the global scale (Shen et al., 2023) have consistently found an underestimation in bottom-up inventories when compared to measurement-based inventories, pointing to a need for improvements in the bottom-up methane inventory methodologies. Furthermore, satellite-based quantification of regional, national, and global methane emissions has emerged as a crucial tool for assessing the accuracies of methane inventories (Jacob et al., 2022; Shen et al., 2023). However, when Bayesian inversion models are used for methane flux quantification, spatially explicit methane inventories are needed as a priori information (Shen et al., 2021, 2023). Past efforts have produced such a priori information by spatially disaggregating methane emissions inventories reported to the UNFCCC (Maasakkers et al., 2023; Scarpelli et al., 2022; EDGAR, 2023), which, as noted above, can have large underestimations and uncertainties in both the magnitude and spatial distributions of oil and gas methane emissions.

In this work, we utilize previous peer-reviewed facility-level measurement data ($n = 1540$) for methane emissions at oil and gas facilities in the major US oil and gas production basins to develop an improved assessment of national, basin-level, and facility-level methane emissions based on oil and gas activity in 2021. Our measurement-based inventory differs from other bottom-up inventories that use generic emission factors (e.g., EPA GHGI) in that we leverage empirical observations to derive insights on facility-level methane emissions distributions that are useful for estimating population mean total methane emissions. Our contributions are threefold. First, we develop statistically robust facility-level methane emission models based on measurement data collected in the years post-2011 (when the EPA's New Source Performance Standards for the oil and gas industry were first proposed) through 2020. We use these models to estimate national methane emissions on both an absolute basis ($\mathrm{Tg\,yr^{-1}}$) and a production-normalized basis (% emitted relative to methane production). Second, we extend this approach to assess the variability and underlying drivers of oil and gas methane emissions and methane loss rates across the major US oil and gas basins. As part of this assessment, we present and compare the quantification of total area methane emissions and the relative contributions of diffuse area emissions versus large concentrated methane sources in the Permian and Uinta basins based on new remote-sensing measurements by MethaneAIR (Staebell et al., 2021; Chulakadabba et al., 2023; Chan Miller et al., 2023), an airborne precursor to MethaneSAT (http://www.methanesat.org, last access: 22 August 2024). Finally, we construct a high-resolution spatially explicit oil and gas methane emissions inventory for 2021, aggregated at the $0.1° \times 0.1°$ ($\sim 10\,\mathrm{km} \times 10\,\mathrm{km}$) spatial scale, and use these results to characterize the spatial patterns in national emissions.

# 2 Methods

## 2.1 Oil and gas activity data

We follow the procedure developed by Omara et al. (2022) to assess the total number and site-level production characteristics of actively producing onshore oil and gas well sites in the US in 2021. The aggregation of wellhead data to well site data (a well site can have multiple wellheads) is needed because (i) methane measurement-based data are reported at the well site level and (ii) production data are reported on a monthly basis for each producing wellhead. Briefly, we use the monthly well-level oil and gas production data as reported by Enverus Prism (Enverus, 2024), which aggregates public and proprietary data on monthly well-level production. For each actively producing well, we derive average well-level oil (in barrels per day, bpd; 1 barrel of crude oil $\sim 0.136\,\mathrm{t}$), gas (in $1000\,\mathrm{ft^3\,d^{-1}}$, Mcfd; $1\,\mathrm{ft^3} = 0.0283\,\mathrm{m^3}$), and combined oil and gas (in barrels of oil equivalent per day, boed; 1 boed = 6 Mcfd gas) production rates based on the reported number of production days in the year and assuming 365 calendar days in the calendar year if the production days were not reported, which occurred at $< 5\%$ of the producing wells (Fig. S10 in the Supplement). We filtered the data for only the actively producing wells ($n = 824\,003$) and used geospatial clustering approaches, described in detail in Omara et al. (2022), to derive well site attributes (e.g., number of wells per site, site-level oil, gas, and boed production). Based on this analysis, we estimate a total of

660 149 actively producing onshore oil and gas well sites in the US in 2021 (Table 1), indicating an average of 1.2 wellheads per well site. Finally, we differentiate between low ($\leq$ 15 boed) and non-low (> 15 boed) production oil and gas well sites based on their average site-level boed production rates in 2021. Our assessment indicates that low production well sites accounted for 82 % of the total number of US onshore actively producing well sites in 2021 (Table 1). We consider these spatial data as comprehensive for the US oil and gas production well sites as it is consistent with the official gross oil and gas production reported by the US Energy Information Administration for 2021 (e.g., the sum of the gross gas production from spatially explicit well-level production data from Enverus Prism is consistent with the total of $\sim 42 \times 10^{12}$ ft$^3$ (trillion cubic feet, Tcf) CE2 of US natural gas gross withdrawals reported by the US Energy Information Administration, https://www.eia.gov/dnav/ng/ng_prod_sum_dc_NUS_mmcf_a.htm, last access: 22 August 2024).

We estimate the total number of operational gathering and transmission compressor stations, natural-gas processing plants, and crude-oil refineries based on spatial data reported by Enverus Prism (Enverus, 2024) supplemented with additional spatial data from the Oil and Gas Infrastructure Mapping (OGIM) database (Omara et al., 2023), which consolidates public-domain data on oil and gas infrastructure locations and facility characteristics. For gathering and transmission pipelines, we estimate total pipeline miles based on the Enverus Prism (Enverus, 2024). In addition, we assess methane emissions associated with gas flaring activity, leveraging the natural-gas flaring detections dataset based on VIIRS (Visible Infrared Imaging Radiometer Suite) instruments onboard the Suomi National Polar-orbiting Partnership (NPP) and NOAA-20 satellites to identify the locations of gas flaring facilities or clusters of facilities and associated gas flared volumes (Elvidge et al., 2015). Table 1 shows the summary statistics for the oil and gas activity data used in this study.

## 2.2 Facility-level measurement-based methane emissions data

We begin by performing a comprehensive data review and assessment of previously published peer-reviewed data on facility-level methane emissions measurements for US oil and gas basins, leveraging Google Scholar search results based on keywords that reflect the geography of interest (oil and natural-gas methane emissions in the US), measurement methods (ground-based facility-level methods, OTM-33A, tracer flux, and mobile transects), and major oil and natural-gas facility categories (production well sites, natural-gas gathering and transmission compressor stations, processing facilities, pipelines, and crude-oil refineries). We focus on ground-based measurement studies that report total facility-level methane emissions quantification for well sites, natural-gas gathering and boosting compressor stations, natural-gas transmission compressor stations, and natural-gas processing plants.

These ground-based measurement approaches include dual-tracer downwind mobile measurements (Mitchell et al., 2015; Omara et al., 2016, 2018), EPA Other Test Method (OTM-33A) downwind stationary measurements (Brantley et al., 2014; Robertson et al., 2017, 2020), and downwind mobile measurements with Gaussian-plume transport modeling (Caulton et al., 2019; Omara et al., 2018). Omara et al. (2018) provide a detailed overview of these ground-based measurement methods. Other recently published studies that used chamber flux quantification approaches and reported only wellhead methane emissions quantification (e.g., wellhead methane emissions in Deighton et al. (2020) and Riddick et al. (2019)) are not included, as unquantified methane sources (e.g., from separators, tanks, and pneumatic devices) likely lead to a low bias in facility-level total methane emissions. However, we use the facility-level total methane emissions data reported by Zimmerle et al. (2020) for natural-gas gathering and boosting stations, based on the aggregation of each facility's onsite component-level measurements performed using a high-flow sampler following leak detection with an infrared camera. We acknowledge a possible low bias in this dataset given the limitations of facility-level measurements using high-flow samplers, including an inability to access all methane-emitting sources and/or to quantify large emission sources beyond the high-flow sampler capacity (Zimmerle et al., 2020). Finally, given their large size and the difficulty of quantifying facility-wide emissions with ground-based measurement approaches, we use available measurement-based methane emissions data for crude-oil refineries based on aerial remote-sensing methods (Lavoie et al., 2017; Duren et al., 2019).

For non-low production well sites, we use previously published facility-level measurement data collected in eight US basins, including the Barnett ($n = 254$; Brantley et al., 2014; Lan et al., 2015; Rella et al., 2015; Yacovitch et al., 2015), Denver-Julesburg ($n = 46$; Robertson et al., 2017; Brantley et al., 2014; Omara et al., 2018), Eagle Ford ($n = 3$; Brantley et al., 2014); Fayetteville ($n = 47$; Robertson et al., 2017), Marcellus Shale ($n = 572$; Omara et al., 2016, 2018; Caulton et al., 2019), Permian ($n = 72$; Robertson et al., 2020), Uinta ($n = 31$; Robertson et al., 2017; Omara et al., 2018), and Upper Green River ($n = 129$; Brantley et al., 2014; Robertson et al., 2017). The consolidated site-level measurement data include data collected in the years post-2011 (when EPA's New Source Performance Standards for the oil and gas industry were first proposed) through 2020. We only focus on data from studies that reported total facility-level emissions quantification in addition to the production characteristics (i.e., gas and/or oil production rates). We use each study's reported facility-level methane loss rate, computed as the methane emissions relative to the methane production at each facility, in our modeling of methane emissions. Where methane loss rates were not reported, we compute the percent methane

**Table 1.** Oil and gas activity data and estimates of total methane emissions.

| Facility category | Facility sub-category | Units | Activity data | Measurement-based data sources (sample size)[a] | Estimated total methane emissions, 2021 (Tg, 95 % CI) | EPA GHGI, 2020 (Tg)[b] |
|---|---|---|---|---|---|---|
| Well sites | Low production | No.CE3 of well sites | 541 987 | $n = 1153$, see footnote for study references | 4.3 (2.9–6.0) | 3.4 |
| | Non-low production | No. of well sites | 118 162 | | 5.1 (3.6–7.4) | |
| Natural-gas compressor stations | Gathering and boosting stations | No. of stations | 4651 | $n = 116$ (Mitchell et al., 2015), $n = 180$ (Zimmerle et al., 2020) | 1.6 (0.9–3.0) | 1.4 |
| | Transmission stations | No. of stations | 2107 | $n = 47$ (Subramanian et al., 2015) | 1.7 (0.7–4.5) | 1.6 |
| Natural-gas processing plants | – | No. of plants | 908 | $n = 16$ (Mitchell et al., 2015) | 1.6 (0.7–3.7) | 0.51 |
| Crude-oil refineries[b] | – | No. of refineries | 143 | $n = 28$ (see footnote) | 0.14 (0.1–0.18) | 0.03 |
| Pipelines | Gathering pipelines | Pipeline miles | 367 717 | EPA Greenhouse Gas Inventory (EPA, 2022) | 0.13 (0.12–0.14) | 0.13 |
| | Transmission pipelines | Pipeline miles | 552 150 | | 0.47 (0.46–0.48) | 0.17 |
| Natural-gas flaring detections | No. of flaring detections | No. of detections | 3153 | $n = 3153$ (Elvidge et al., 2015) | 0.56 (0.55–0.57) | – |
| | Estimated gas flared volumes | MMcf yr$^{-1}$ | 344 217 | Elvidge et al. (2015) | | |
| | | | | Total estimated methane emissions | 15.7 (14.1–18.0) | 8.3 (7.0–9.6) |

[a] Measurements at well sites include 1153 facility-level measurements from nine studies in eight basins or production regions in the US. Studies include Brantley et al. (2014), Robertson et al. (2017, 2020), Omara et al. (2016, 2018), Caulton et al. (2019), Rella et al. (2015), Lan et al. (2015), and Yacovitch et al. (2015). For crude-oil refineries, available facility-level measurements are based on aerial remote-sensing quantification (Duren et al., 2019; Lavoie et al., 2017). [b] The EPA GHGI total includes 0.5 Tg methane from natural-gas distribution, liquified-natural-gasCE4 storage, and other sources not shown in this table. MMcf: $1 \times 10^6$ ft$^3$ (1 ft$^3$ = 0.0283 m$^3$).CE5

loss rates as follows, based on the reported average gas production rate at the time of measurement:

$$\text{Methane loss rate [unitless]} = \text{CH}_4 \left[ \frac{\text{kg}}{\text{h}} \right] \times \frac{1}{\text{Gas [Mcfd]}}$$
$$\times \frac{1 \, \text{Mcf}}{19.2 \, [\text{kgCH}_4]} \times \frac{1}{\sigma_{\text{CH}_4}} \times \frac{24 \, \text{h}}{1 \, \text{d}}, \quad (1)$$

where CH$_4$ [kg h$^{-1}$] is the measured facility-level methane emission rate in kg h$^{-1}$, Gas[Mcfd] is the reported gas production rate in Mcfd, Mcf is 1000 ft$^3$, 19.2 kg/Mcf is the methane density at 60 °F (15.5 °C) and 1 atm, and $\sigma_{\text{CH}_4}$ is the assumed methane content of the produced natural gas (we assume an average of 80 % methane content in the produced natural gas).

For low production well sites ($\leq 15$ boed), we use the same facility-level methane emissions data and emissions assessment methods as described in detail in Omara et al. (2022). Briefly, we use the reported empirical observations ($n = 240$; Omara et al., 2022) in a hybrid Monte Carlo and non-parametric probabilistic model that simultaneously estimates the frequency of below-detection-limit sites, the frequency of high-emitting sites representing the top 5 % of emitting facilities based on absolute methane emissions, and the distribution of high-emitter methane emissions while accounting for the weakly observed positive relationship between emission rates and production rates for the bottom 95 % of emitting well sites. We integrate this model with spatially explicit activity data on low production oil and gas well sites in 2021 (Enverus, 2024) to estimate their total methane emissions.

For non-low production well sites ($> 15$ boed), we use the reported site-level measurement data described above and shown in Fig. 1a, which indicates an inverse relationship between production-normalized methane loss rates and facility-level gas production rates (Omara et al., 2018). The measurement-based data include measurements that were reported as zeros or were below the method detection limits of 0.036 kg h$^{-1}$ (Robertson et al., 2017; Brantley et al., 2014)

for the OTM-33A methods and $0.01 \, \text{kg} \, \text{h}^{-1}$ (Omara et al., 2016) for the dual-tracer flux quantification.

Figure 1b shows the previously reported facility-level measurements at midstream/downstream facilities, including natural-gas gathering and boosting compressor stations, transmission compressor stations, processing plants, and crude-oil refineries. In all cases, we use the average facility-level methane emissions data as reported, acknowledging that inherent limitations in these measurement approaches (e.g., pseudo-random facility-level measurements with small sample sizes in ground-based approaches or difficulty quantifying large emitters using high-flow samplers in component-level measurements) likely increase the uncertainties in our estimates of total, regional, and national methane emissions.

## 2.3 Facility-level methane emissions model development and estimation of total national methane emissions

Our approach for estimating regional and national oil and gas methane emissions builds upon previous works that used data from hundreds to thousands of ground-based facility-level measurements (Zavala-Araiza et al., 2015; Alvarez et al., 2018; Omara et al., 2018, 2022) in combination with robust probabilistic models integrated with oil and gas activity data. Zavala-Araiza et al. (2015) and Alvarez et al. (2018) demonstrated that measurement-based inventories developed using these methods produce total methane emission results that are in good agreement, within statistical uncertainty, of independent airborne measurements of total area methane emissions.

For non-low production well sites (average facility-level production rates > 15 boed), we begin by evaluating facility representativeness on the basis of (i) the geographical diversity of measurements, (ii) the distribution of facility-level production rates of measurements compared with the national population of well site facilities, and (iii) the distribution of facility-level methane emission rates across basins (Fig. S3). Our measurement data, while limited in sample size, cover eight major US oil and gas basins with diverse oil and gas production characteristics, including the Appalachian, Permian, Uinta, Barnett, Fayetteville, Greater Green River, and Denver-Julesburg. The wide range of basin-level gas-to-oil ratios ($\sim 1$ to 800 Mcf/barrel) is well represented in the data for the sampled basins (Fig. S3b).

In addition, the distribution of facility-level natural-gas production rates shows reasonable overlap with that for the national population of non-low production facilities, and the broad range in distribution of facility-level production rates across the national population of sites ($\sim 90$ to > 50 000 Mcfd) is well represented in the sampled sites (Fig. S3c). However, the distribution of production rates for the sampled sites suggests potential bias toward higher-producing sites relative to the national distribution (Fig. S3c). We account for any such potential biases by developing emissions models based on production-normalized methane loss rate distributions (methane emitted relative to methane produced) across seven cohorts of specific gas production rates (further details below).

We develop and use probabilistic emission rate distributions based on production-normalized methane loss rates, which show a wide range ($< 0.01 \%$ to > 90 %; Fig. 1a) across all basins (Fig. S3d), reflecting, in part, the diversity in production characteristics within and across basins. We use production-normalized methane loss rate distributions because (i) the empirical data across a wide diversity of oil and gas production facilities suggest an inverse relationship in which high-producing facilities exhibit comparatively low methane loss rates, and vice versa (Fig. 1a), and (ii) the consolidated dataset includes measurements collected in earlier years (before 2021). By using the production-normalized methane loss rate distribution models for specific cohorts of facility-level production rates, we do not model any particular site that is active in 2021 as exhibiting the same emission rate size as observed when measurements were taken in the past, as the empirical data and the model constrain facility-level methane loss rates to production levels, which will be time variant. As such, we provide a necessary constraint on our estimates, effectively adjusting modeled facility-level methane emission rates if production rates have substantially changed over time.

To estimate regional methane emissions for non-low production well sites, we group the data for the empirical facility-level methane loss rates into seven log-normalized gas production (Mcfd) cohorts, as shown in Fig. 1a and delineated by dashed vertical lines (log-Mcfd $\leq$ 5, 5–6, 6–7, 7–8, 8–9, 9–10, and log-Mcfd > 10). We use one log-e space (between log-Mcfd $\leq$ 5 and log-Mcfd > 10) to develop these production cohorts, given the inverse relationship between facility-level methane loss rate and production rates, and they are selected to provide sufficient sample sizes for emissions distribution modeling for each production cohort (Fig. 1a). For each cohort, we simulate the frequency of finding a site emitting below the method detection limits (reported as zeros or below the method detection limit) through a random bootstrapping procedure, repeated $10^4$ times, with replacement. From this simulation, we develop a frequency distribution for the sites below the detection limits ($f_{BDL}$), which averaged roughly 20 % to 30 % for all cohorts with the exception of the last production cohort ($> 10$ Mcfd), where the frequency drops to roughly 10 % to 20 % (Fig. S1).

For the measured oil and gas production well sites with emissions above the method detection limits, we begin by applying a log transformation to the reported facility-level methane loss rates in each cohort and assessing the goodness of fit of the empirical distributions to a lognormal distribution, using the Kolmogorov–Smirnov test with significance established at $p < 0.05$. For all seven cohorts, we find that the lognormal distribution assumption is valid, with $p > 0.05$ (Fig. S2). For each cohort's empirical distribution,

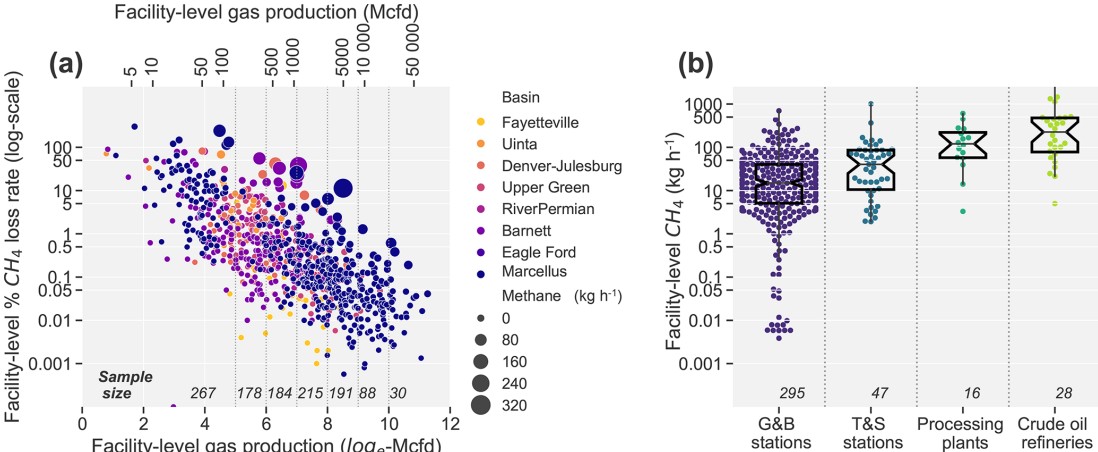

**Figure 1.** Previously reported facility-level measurement-based methane emissions data. **(a)** Facility-level methane emissions data (percent methane loss rate) as a function of gas production rate ($n = 961$ non-low production well sites). The bottom $x$ axis shows the log-normalized gas production rates, with dashed vertical lines delineating the seven production cohorts used to model total methane emissions. Sample sizes for the production cohorts are shown at the bottom of the plot, above the bottom $x$-axis tick labels. The top $x$ axis shows the same production data in Mcfd. Each point is color coded by basin and sized in proportion to the quantified methane emission rate in $kg\,h^{-1}$. Measurements that were reported as being below the method detection limits are not shown. **(b)** Absolute methane emission rate data ($kg\,h^{-1}$) for gathering and boosting (G&B) compressor stations ($n = 295$), transmission and storage (T&S) compressor stations ($n = 47$), natural-gas processing plants ($n = 16$), and crude-oil refineries ($n = 28$). The swarm plots show individual facility-level measurements, while the notched box plots show the distributions (the boxes represent the 25th and 75th percentiles and the whiskers extend to $1.5 \times$ the interquartile range).

we assume a univariate normal likelihood with mean $\mu$ and standard deviation $\sigma$ and use Bayesian models with weakly informative priors to estimate $\mu$ and $\sigma$, for example, as $\mu \sim \text{Normal}(-10, 5)$ and $\sigma \sim \text{HalfNormal}(3)$ for the first co-
5 hort of non-low production sites. For Bayesian inference, we draw 5000 posterior samples from the posterior distribution using the PyMC3 (Salvatier et al., 2016) implementation of the No-U-Turn Sampler (NUTS) algorithm (Hoffman and Gelman, 2014), from which we estimate $\mu$ and $\sigma$ as well as
10 the 94 % highest posterior density (HPD) intervals. Note that the mean facility-level methane loss rate for each cohort can be computed as $\exp(\mu + 0.5\sigma^2)$. From the posterior results, we generate 5000 predictions of the facility-level methane loss rate for each measured well site within each production
15 cohort. Figure 2 shows the cumulative probability distribution function for the observed data and 500 random samples from the model predictions.

We follow a similar Bayesian modeling procedure to develop predictions of emissions distributions ($kg\,h^{-1}$ per fa-
20 cility), conditional on empirical data, for the gathering and boosting compressor stations, transmission compressor stations, natural-gas processing plants, and crude-oil refineries. For these facility categories, we use the measured mean absolute methane emissions data as is ($kg\,h^{-1}$ per facility) in
25 our models, as we lack natural-gas capacity or throughput information for the national population of facilities.

We then proceed as follows to estimate methane emissions for the total population of facilities. For every facility in each facility category and/or production cohort, we randomly

draw an emission rate from the modeled posterior predictions 30 (Fig. 2). For non-low production oil and gas production facilities, we randomly draw a methane loss rate estimate which is then multiplied by the facility's average methane production rate to estimate methane emissions in $kg\,h^{-1}$. As some facilities can have emissions below the method detection lim- 35 its, we decrement the total estimated emission rate based on the randomly sampled frequency of sites below the detection limit ($f_{\text{BDL}}$) randomly drawn from the modeled distributions. We repeat this procedure 500 times and develop a methane emissions distribution for the total methane emis- 40 sions for each facility category or production cohort.

Given the scarcity of facility-level measurements for gathering and transmission pipelines, we use the emission factors estimated by the US EPA Greenhouse Gas Emission Inventory (177 and 362 kg methane $km^{-1}\,yr^{-1}$, respectively; EPA, 45 2022) and assume normal distributions of emission factors with 50 % uncertainty. Our use of the EPA's GHGI emission factors for these emission sources makes it possible to provide a more complete spatially explicit inventory of oil and gas methane emissions (inclusive of gathering and transmis- 50 sion pipelines for which we have geospatial activity data), but it likely increases uncertainties in our total methane emission estimates given the potential underestimation in the GHGI emission factors.

We also estimate the methane emissions associated with 55 gas flaring activities using location-specific gas flaring data from the VIIRS instrument (Elvidge et al., 2015) and apply an average effective methane destruction removal efficiency

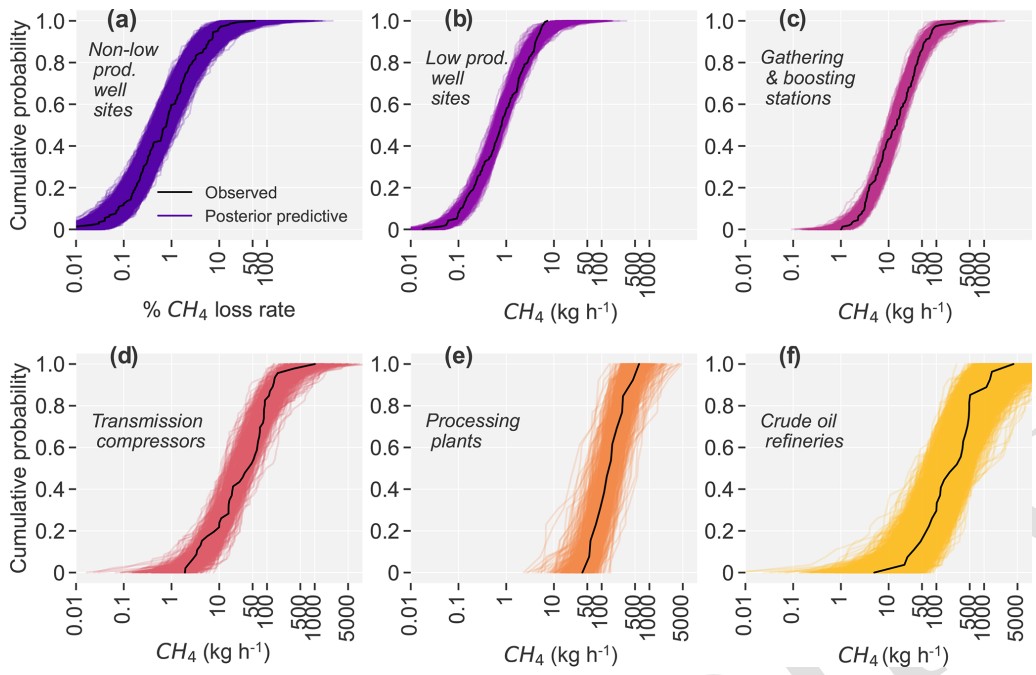

**Figure 2.** Empirical cumulative distribution functions of observed data and model predictions. Empirical CDFs are shown as solid black lines while thin colored lines show 500 random samples drawn from the model predictions. Sample sizes and data sources for empirical data are shown in Table 1. **(a)** Non-low production well sites (modeled as facility-level methane loss rates); the plot shows the CDF for the $5 < \log$-Mcfd $< 6$ production cohort. Figure S2 shows the CDFs for all seven non-low production cohorts in Fig. 1. **(b)** Low production well sites (kg h$^{-1}$ per site). **(c)** Gathering and boosting compressor stations. **(d)** Transmission compressor stations. **(e)** Natural-gas processing plants. **(f)** Crude-oil refineries.

of 91 % (95 % confidence interval of $\sim$ 90 %–92 %; Plant et al., 2022).

Finally, we combine the emissions distributions for all facility categories and sources using Monte Carlo methods to estimate the mean total national methane emissions and the 95 % confidence interval based on the 2.5th and the 97.5th percentiles of the modeled distributions. Figure 3 shows a general schematic of the emissions model development and the estimation of total methane emissions.

## 2.4 Spatial allocation of estimated methane emissions and basin-level methane loss rates

For each facility with a known location (latitude, longitude), our assessment includes 500 different estimates of likely facility-level methane emission rates (in kg h$^{-1}$), from which we derive 500 different estimates of total national methane emissions. We use 500 simulation results for each facility as a reasonable simulation size that is not too computationally intensive to implement but also gives sufficient statistical power to develop a robust model uncertainty assessment. We use a search algorithm to identify a random sample of the facility-level emission rate distribution that most closely matches the computed mean estimate for the population of facilities. We use a similar approach to select a random sample of the facility-level emissions distributions rep-

resenting uncertainties in the total emission estimates (i.e., the distribution that most closely matches the lower bound and upper bound of the 95 % confidence interval for the total estimated methane emissions). We then aggregate the total mean methane emissions (and the associated upper- and lower-bound estimates) on regular grids of 0.1° × 0.1° decimal degrees ($\sim$ 10 km × 10 km) to produce spatially explicit oil and gas methane emissions inventories and related uncertainties in the total methane emissions within each grid.

Our spatial allocation of estimated total oil and gas methane emissions is dependent, in part, on the completeness and spatial accuracy of the oil and gas infrastructure locations for specific regions and oil and gas basins, for which related uncertainties are difficult to quantify based on available information. Our spatial allocation provides the mean methane emissions estimates for the year 2021 aggregated in each 0.1° × 0.1° grid cell ($\sim$ 10 km × 10 km) and are not intended to characterize methane emissions at a specific point in time, as substantial short-term variability in emissions may occur due in part to the stochastic character of facility-level methane emissions.

We compute basin-level and national methane loss rates as the ratio of estimated basin-level methane emissions to the gross methane production in 2021, based on gross natural-gas production data from Enverus Prism (Enverus, 2024) and an assumed average methane content of 80 % in natural gas.

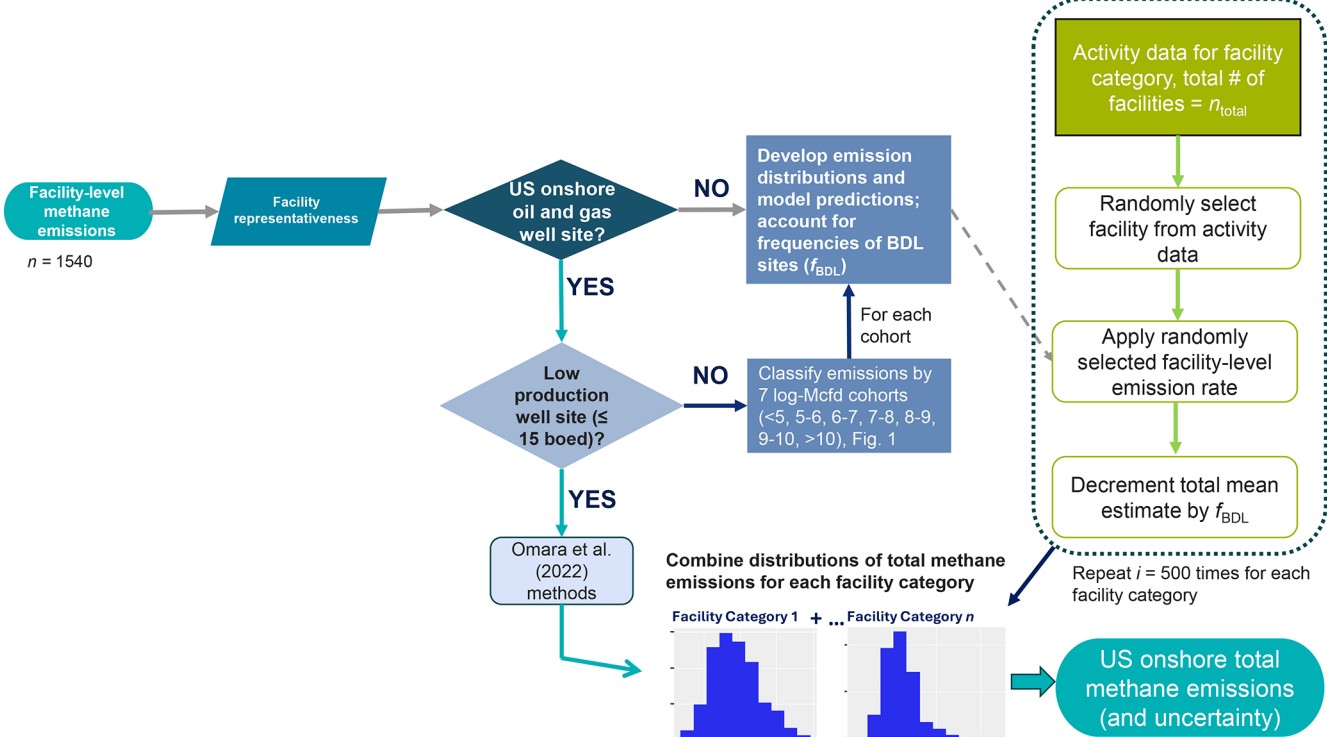

**Figure 3.** General schematic for model development and the estimation of total methane emissions, given activity data for each facility category.

Our assumption of an average 80 % methane content in natural gas is informed by regional estimates of methane composition in natural gas based on the EPA GHGI (EPA, 2022). We acknowledge that uncertainties in methane composition across basins likely increase uncertainties in our overall methane loss rate calculations. Further studies on basin-level methane composition are needed to constrain these uncertainties. This methane intensity metric allows for a direct comparison of estimated methane losses relative to gross methane production across different basins. While our use of gross methane production accounts for emissions from associated gas produced during oil operations, the results are not intended to represent life-cycle emission intensities, which are outside the scope of this work.

## 2.5 Model uncertainties and limitations

In our modeling, we use the average facility-level emissions data as is while assuming that facility emissions arise from an underlying methane emissions distribution that is statistically described by lognormal distributions. The implementation of these probabilistic models produces emissions distribution models (Fig. 2) that account for uncertainties in each facility's measured average methane emission rate and the facility-to-facility variability in methane emissions within and across multiple oil and gas production regions. The 95 % confidence intervals obtained through the Monte Carlo methods above reflect these uncertainties as well as the model uncertainties in the predictions of emissions distributions, given the limited sample sizes used herein. Additional uncertainties that are difficult to quantify include uncertainties in the oil and gas activity data and uncertainties in the potential impacts of recently promulgated federal/state-specific regulations or operator-specific practices regarding regular facility-level methane emissions monitoring and repair. In addition, due to a lack of comprehensive spatially explicit activity data, our measurement-based inventory does not include methane emissions from downstream natural-gas distribution, liquified-natural-gas storage, post-meter emissions, and abandoned oil and gas wells. The EPA GHGI (EPA, 2022) estimates that these sources account for $\sim 0.5$ to $1\,\mathrm{Tg\,yr^{-1}}$ of the total methane emissions; the vast majority of these would be distributed in urban locations outside of major oil and gas production regions.

## 3 Results and discussion

### 3.1 Total national oil and gas supply chain methane emissions

We estimate a measurement-based methane emissions inventory (EI-ME) of total national oil and gas methane emissions for the onshore US of 15.7 Tg (95 % confidence interval of 14–18 Tg or $-10\,\%/+15\,\%$ uncertainty; Table 1;

Fig. 4) for the year 2021. Our central estimate and confidence bounds are in reasonable agreement with recent measurement-based facility-level emission estimates (Alvarez et al., 2018; Rutherford et al., 2021 (production sector only)) and satellite-derived oil and gas methane emissions, including quantifications using GOSAT (Lu et al., 2022, 2023) and TROPOMI (Shen et al., 2022) (Fig. 3b). In addition, consistent with previous findings (Alvarez et al., 2018; Rutherford et al., 2021; Shen et al., 2022), our central estimate is significantly greater than inventories developed using the traditional bottom-up source-level emission factor approaches: we find $1.9\times$ and $1.8\times$ greater total methane emissions than are estimated by the EPA Greenhouse Gas Inventory (EPA, 2022) and EDGAR v8 (EDGAR, 2023) inventories for the year 2021, respectively (Fig. 4).

We attribute the largest discrepancy between our measurement-based estimates and the EPA GHGI to the estimated emissions for the oil and gas production sector, which we estimate accounts for approximately 60 % of the total onshore methane emissions – a total of $\sim 9$ Tg in 2021, roughly $2.6\times$ greater than the EPA GHGI's estimate for the production-related methane emissions (Fig. 4; Table 1). These results are in reasonable agreement with previous measurement-based inventories (Alvarez et al., 2018; Rutherford et al., 2021; Omara et al., 2022) and, as has been noted elsewhere (Alvarez et al., 2018; Rutherford et al., 2021; Omara et al., 2018), likely reflect the use of emission factors in the EPA GHGI that do not adequately characterize the contributions of high-emitting methane sources that have been consistently observed in measurement-based studies. Furthermore, within the oil and gas production sector, we find that the low production well site cohort ($< 15$ boed) accounts for roughly one-half of the total production site methane emissions in 2021, consistent with recent findings based on 2019 oil and gas activity (Omara et al., 2022). As Table 1 shows, the estimated total methane emissions from the low production well site cohort alone are $\sim 26$ % more than the total methane emissions from all low production and non-low production well sites based on the EPA GHGI.

In 2021, we estimate a national methane loss rate of 2.6 % (95 % CI: 2.3 %–2.9 %) relative to gross natural-gas production, assuming an average of 80 % methane content in natural gas (see the Methods section). Our average methane loss rate assessment is in reasonable agreement with recent satellite-derived estimates (Shen et al. (2022) using TROPOMI and Lu et al. (2023) using GOSAT). Lu et al. (2023) report a steadily declining national methane loss rate between 2010 ($\sim 3.7$ %) and 2019 ($\sim 2.5$ %) and attribute these trends to two likely factors: (i) a slower increase or decrease in absolute methane emissions compared to the increase in methane production during this period and (ii) the impact of national regulations, such as the EPA's New Source Performance Standards, promulgated in 2012, which focused on reducing emissions from newly constructed well sites, among other requirements. As we discuss further below, we find significant variability in the total methane emissions as well as the spatial distributions of the estimated emissions at the regional/basin level for the oil and gas activity in 2021.

## 3.2 Variability in estimated basin-level methane emissions

Among the major oil and gas production basins, we identify the Permian, Appalachian, Anadarko, Eagle Ford, Haynesville, and Barnett basins as the top six methane-emitting basins, with estimated mean total basin-level methane emissions ranging from approximately 70 to 340 t h$^{-1}$ (Table 2, Fig. 5). These six basins account for 72 % of the onshore total combined oil and gas production (boed) and 52 % of the estimated total oil and gas methane emissions. Among these basins, we estimate considerable variability in gas production-normalized methane loss rates, with the lowest mean methane loss rates of $< 1$ % occurring in the Appalachian and the Haynesville basins and the highest mean methane loss rates of 3 %–4 % occurring in the Permian, Anadarko, and Barnett (Table 2). The basin-level differences in methane loss rates among basins are consistent with the GOSAT-derived estimates for 2019 (Lu et al., 2023, Table 2) except for the Appalachian and the Eagle Ford basins, where this study's estimates are roughly $2\times$ greater (Table 2). As with our findings on the comparative assessments with the EPA GHGI at the national level, our basin-level methane emission estimates are consistently greater than the EPA GHGI estimates (Maasakkers et al., 2023) by factors of $1.7\times$ (Appalachian) to $\sim 4\times$ (Anadarko).

The confluence of various possible factors, including the spatial density and characteristics of methane-emitting oil and gas infrastructure and basin-level operational characteristics (gas dominant versus oil dominant, intensive flaring versus basins with negligible flaring, etc.), contribute to the differences in the modeled basin-level methane emissions. In each basin, we estimate a predominant contribution of total methane emissions from well site infrastructure, ranging from 55 % to 75 % of the total basin-level methane emissions (Table 2; Fig. 5). Well site infrastructure characteristics vary significantly among basins; for example, the Appalachian Basin is characterized by a large population of old, leak-prone, low-producing gas well sites (Omara et al., 2016; Deighton et al., 2020; Riddick et al., 2019), although more than 95 % of the gas produced there comes from the $\sim 3$ % of well sites that are unconventional non-low production well sites (Enverus, 2024). This contrasts with the San Joaquin Basin, where well site infrastructure is dominated by low-producing oil pump jacks with limited onsite processing equipment, which in turn contrasts with the oil-dominant Bakken, dominated by high-producing horizontally drilled well site facilities that typically have multiple wellheads and auxiliary processing equipment including separators, storage tanks, and flare stacks. These varying basin-level oil and gas infrastructure characteristics likely contribute to the modeled

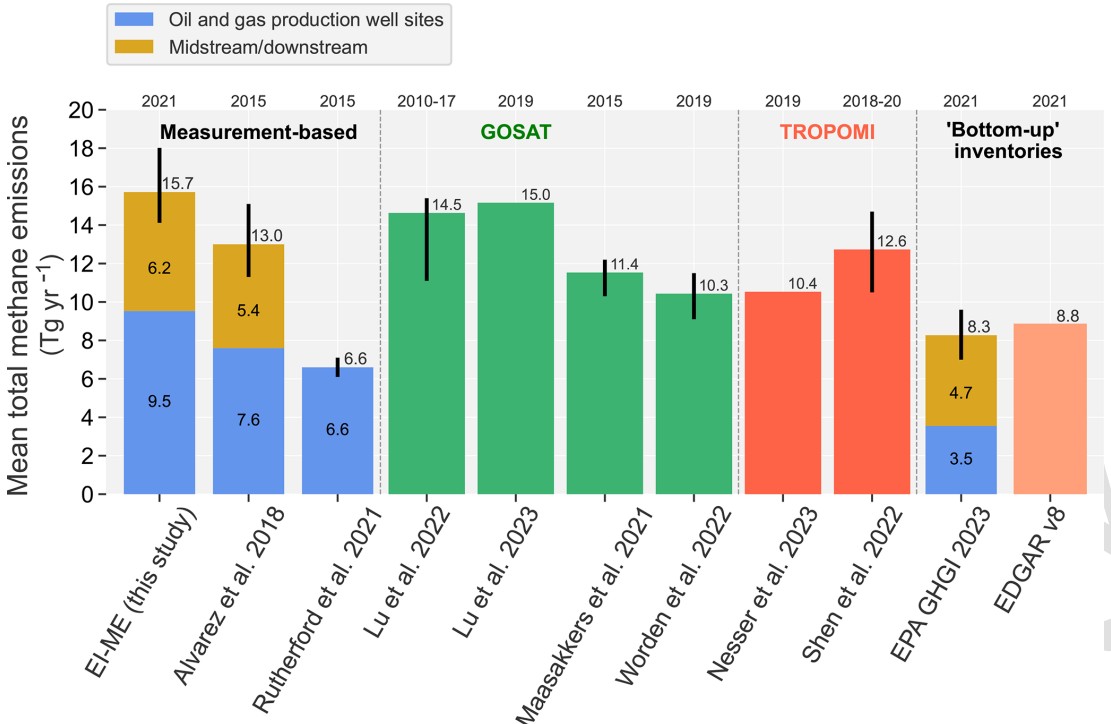

**Figure 4.** Comparison of this study's national estimate of total methane emissions from the oil and gas supply chain with previous measurement-based estimates. The first three bars show the oil and gas methane emissions estimated from facility-level measurements (this study; Alvarez et al., 2018) and production-sector-only methane emissions estimated by Rutherford et al. (2021) using models developed from component-level measurement data. Blue bars show the estimated emissions for the production sector. Gold bars show the estimated emissions for the midstream and downstream facilities (compressor stations, processing plants, refineries, and gathering and transmission pipelines). Error bars show the estimated 95 % confidence bounds on the mean total methane emission estimates. This study's estimate of total national methane emissions includes $\sim 0.1\,\mathrm{Tg\,yr^{-1}}$ of estimated methane emissions for Alaska. The green bars and the red bars show the satellite-derived estimates for the contiguous US based on GOSAT and TROPOMI observations, respectively. The last two bars show the bottom-up inventories from EPA GHGI and EDGAR v8 for the contiguous US. In all cases, the years for which methane emissions are estimated are shown on the top $x$ axis.

emission differences, given that the empirical data synthesized herein reveal the weak correlation of methane emission profiles with well site production characteristics (loss rates; Fig. 1a) and infrastructure category (absolute emissions; Figs. 1b, 2).

Furthermore, the magnitude of modeled methane emissions varies by basin-level operational characteristics. For example, the Permian Basin, with its significant new oil and gas development, stands in contrast to the relatively mature basins such as the Barnett or Uinta with steadily declining gas production and aging well site infrastructure. As Lu et al. (2023) observed, high methane loss rates tend to be associated with oil-dominant basins (e.g., the Permian, Eagle Ford, and Bakken), where production activities are focused on oil production, even though substantial associated gas is co-produced along with the oil. In these basins, potentially higher methane emissions may occur due to venting and/or inefficient flaring of the co-produced gas, especially when there is insufficient infrastructure to gather and process the associated gas production and then transport it to market, as

has been postulated for the Permian Basin (Lyon et al., 2021; Varon et al., 2023; Lu et al., 2023). As noted previously, basin-level differences in total methane emissions could also be impacted by federal/state-level regulations for oil and gas methane emissions and/or operator-specific practices, affecting both the magnitude and temporal variability of emissions. While our methods are based on insights derived from empirical observations and robust modeling to estimate methane emissions specific to oil and gas activity in 2021, we lack sufficient data to characterize the impacts of specific regulations or operator practices. Further studies are needed to assess oil and gas methane emission trends and corresponding underlying drivers.

### 3.3 Sub-basin methane assessment and comparison with emissions quantification using MethaneAIR

This study's EI-ME inventory provides methane emission estimates at geolocated oil and gas facilities, making it possible to develop aggregate methane emission estimates across sub-

**Table 2.** Top six methane-emitting basins: production, loss rate, and comparison with the EPA GHGI (traditional bottom-up inventory) and Lu et al. (2023)[a] (satellite-derived estimates).

| Basin | Basin area (km²) | Well site count (% from low prod.) | Total annual (2021) gas production (Tcf[b]) | EI-ME methane emissions, 2021 (t h⁻¹ (95% CI) \| % from well sites) | EPA GHGI methane emissions, 2020 (t h⁻¹) | EI-ME methane loss rate, 2021 (% (95% CI))[c] | EPA GHGI methane loss rate, 2020 (%) | GOSAT methane loss rate, 2019 (%),[d] from Lu et al. (2023) | TROPOMI methane loss rate, 2019 (%),[e] from Shen et al. (2022) |
|---|---|---|---|---|---|---|---|---|---|
| Permian | 165 325 | 129 364 (78 %) | 6.5 | 335 (274–428) \| 69 % | 106 | 2.6 (2.1–3.3) | 1.0 | 2.7 (1.6–3.0) | 3.5–4.6 |
| Appalachian | 415 446 | 167 132 (97 %) | 12.7 | 231 (165–324) \| 75 % | 145 | 0.92 (0.66–1.30) | 0.68 | 0.45 (0.40–0.47) | 0.46 |
| Anadarko | 42 479 | 24 180 (64 %) | 1.9 | 119 (93–166) \| 55 % | 33 | 3.2 (2.5–4.4) | 1.0 | 3.4 (2.1–3.6) | 1.5 |
| Eagle Ford | 50 179 | 24 377 (54 %) | 2.3 | 90 (73–119) \| 75 % | 29 | 2.0 (1.7–2.7) | 0.69 | 1.1 (0.78–1.3) | 2.0 |
| Haynesville | 28 922 | 23 895 (78 %) | 4.8 | 75 (59–95) \| 69 % | 30 | 0.80 (0.63–1.0) | 0.41 | 1.2 (0.89–1.2) | 1.0 |
| Barnett | 68 146 | 25 760 (79 %) | 0.92 | 74 (57–96) \| 68 % | 35 | 4.1 (3.1–5.5) | 2.0 | 4.0 (3.3–4.1) | 2.6 |

Note the differences in temporal resolution between the studies used for the comparison; specifically, the EPA GHGI basin-level estimates are based on Maasakkers et al. (2023) for the year 2020 (the latest year for which spatially explicit estimates are available), the Lu et al. (2023) GOSAT estimates are for the year 2019, and the Shen et al. (2022) TROPOMI estimates are based on satellite observation data aggregated over the period between May 2018 and February 2020. [a] Loss rates calculated assuming 90 % methane content in each basin for ease of comparison with Lu et al. (2023). [b] $1 \times 10^{12}$ ft³. [c] Methane loss rate calculated using 2021 production data from Enverus Prism. [d] Methane loss rate calculated using 2019 production data from Enverus Prism. [e] For the Permian, Shen et al. (2022) report posterior emissions in the range of 2.9 to 3.7 Tg yr⁻¹, representing a production-normalized methane loss rate of 3.5 % to 4.6 %.

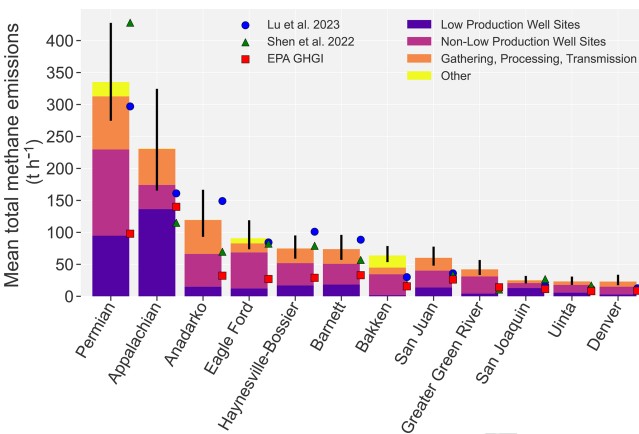

**Figure 5.** Basin-level differences in modeled mean total methane emissions and comparison with the EPA GHGI (Maasakkers et al., 2023), TROPOMI-derived estimates (Shen et al., 2022), and GOSAT-derived estimates (Lu et al., 2023). See Fig. S11 for a similar chart showing the state-by-state breakdown for the top 10 emitting US states based on the EI-ME inventory estimates.

basin to basin and national levels. We compare our sub-basin estimates for the Delaware portion of the Permian Basin and the Uinta Basin with new remote-sensing-based quantification by MethaneAIR (Staebell et al., 2021; Chulakadabba et al., 2023; Chan Miller et al., 2023), an airborne precursor to the MethaneSAT satellite, which was launched in March 2024. The MethaneAIR and MethaneSAT missions are managed by MethaneSAT LLC (http://www.methanesat.org, last access: 22 August 2024), which is a wholly owned subsidiary of Environmental Defense Fund. Both MethaneAIR and MethaneSAT are designed to produce quantitative data on total regional methane emissions while spatially disaggregating diffuse area emissions and detecting high-emitting point sources. Detailed descriptions of the MethaneAIR instrument technical specifications, instrument calibration, retrieval methods, and point-source detections and validation can be found in recent works by Staebell et al. (2021), Conway et al. (2024), Chulakadabba et al. (2023), El Abbadi et al. (2023), Chan Miller et al. (2023), and Omara et al. (2023).

In August 2021, MethaneAIR flew across a $\sim 10\,000\,\mathrm{km}^2$ area in the Delaware Sub-basin of the Permian Basin (research flight RF-06) and the Uinta Basin (research flight RF-08) and produced a quantification of the total area methane emissions using a geostatistical inverse modeling (GIM) framework (based on Miller et al., 2020). The GIM framework was applied to the inversion of the column mean methane dry-air mole fraction retrieved using MethaneAIR measurements obtained while flying at $40\,000\,\mathrm{ft}$ ($\sim 12\,000\,\mathrm{m}$) above the ground aboard the NCAR GV aircraft (https://www.eol.ucar.edu/field_projects/methaneair, last access: 22 August 2024). For MethaneAIR and MethaneSAT, the GIM framework is specialized to exploit the instrument's high spatial resolution, wide spatial

coverage, and high precision. It ingests high-emitting point-source detections, which are quantified using the modified integrated mass enhancement method (Chulakadabba et al., 2023). As such, remote-sensing measurements by MethaneAIR at 40 000 ft ($\sim$ 12 000 m) above the ground produce a high-resolution spatially explicit quantification of the total area methane emissions as well as high-emitting methane point sources emitting above $\sim$ 200 kg h$^{-1}$.

We compare this new MethaneAIR total area methane quantification with the EI-ME modeled results as well as the results from previous peer-reviewed studies in domains overlapping with the RF-06 Permian and the RF-06 Uinta regions. For both regions, we find good agreement, within uncertainty bounds, of the MethaneAIR quantification with other studies (Fig. 6), with emission rate quantifications that fall within representative ranges of mean total sub-basin methane emissions of 80–100 and 17–24 t h$^{-1}$ for RF-06 Permian and RF-08 Uinta, respectively (horizontal dashed lines in Fig. 6).

Based on the MethaneAIR quantifications for these two regions, we estimate that diffuse area emissions (which are assessed using the GIM modeling framework) account for the majority of the methane emissions in both sub-basins; they represent 63 % and 88 % of the total area methane emissions in the RF-06 and RF-08 regions, respectively. The remainder (37 % and 12 % of the total in RF-06 and RF-08, respectively) is attributable to the quantified high-emitting methane point sources with facility-specific methane emission rates in excess of $\sim$ 200 kg h$^{-1}$ per facility. These results are in reasonable agreement with the EI-ME results – averaged over the year – for the same spatial domains, in which oil and gas methane sources with mean methane emission rates $< 200$ kg h$^{-1}$ account for 85 % and 90 % of the total estimated methane emissions for RF-06 and RF-08, respectively. Furthermore, Cusworth et al. (2022) report similar results for the same regions overlapping with these domains in the Permian and Uinta. They find that methane sources below 200 kg h$^{-1}$ account for 70 % and 88 % of total area emissions, which were quantified based on the area inversion of TROPOMI satellite observations and point-source detections by AVIRIS-NG in 2021 and 2020 for RF-06 and RF-08, respectively. Furthermore, for a different sub-region of the Permian Basin, Kunkel et al. (2023) observed that facility-sized emission sources with rates below 280 kg h$^{-1}$ contribute 67 % of the total emission rate from all sources with rates above 10 kg h$^{-1}$. At the national level, Omara et al. (2022) previously showed that the large population of low-producing well sites (also known as marginal wells) with population-average methane emission rates of $\sim$ 1 kg h$^{-1}$ per site account for roughly one-half of all production site methane emissions. Taken together, these results underscore the importance of small methane-emitting sources dispersed across areas; while individually emitting at low rates, in aggregate, these sources can nevertheless contribute a disproportionate fraction of the regional total methane emissions.

Williams et al. (2024) expand on these assessments, providing a detailed look at facility-level methane emissions distributions at the basin and national level.

## 3.4 Variability in estimated spatial distribution of methane emissions

Our spatially explicit EI-ME inventory suggests that basin-level differences also manifest as differences in the spatial distribution of total methane emissions. On average, we find oil and gas methane emission hotspots in every major US oil and gas production basin, including the Permian (the largest oil-producing basin in the US, located in western Texas and southern New Mexico), Appalachian (Pennsylvania, Ohio, West Virginia, and New York), Anadarko (Oklahoma and Texas), Eagle Ford (Texas), Bakken (North Dakota), and Haynesville (Texas and Louisiana; Figs. 6, S4) basins. Our analysis suggests that methane emission hotspots tend to be concentrated in areas with high oil and gas production, as evidenced, for example, by the two large hotspots in the rapidly developing, high-producing Delaware and Midland sub-basins of the Permian Basin (Fig. 7a; see Fig. S9 for maps representing the lower bounds and upper bounds on spatially gridded emissions), consistent with spatial distributions for the satellite-observed methane emissions quantification in this region (Zhang et al., 2020; Varon et al., 2023). In addition, as with the total basin-level emissions, methane emissions spatial distributions are functions of oil and gas activity and their related facility-level emission characteristics. For example, substantial low production oil and gas well site activity yields modeled methane emission hotspots in the southwestern tip of the Appalachian Basin (Fig. 7a), even though this region is not an oil and gas production hotspot (Fig. S5). Furthermore, our analysis suggests the spatial correlation of methane emission hotspots with intensive gas flaring activity, particularly for the oil-producing basins with substantial associated gas production, including the Permian, the Eagle Ford, and the Bakken regions (Figs. 4, S5).

We further assess the variability in the spatial distribution of modeled methane loss rates, which reveals areas ($25 \times 25$ km$^2$ grids) in each major basin where methane loss rates are $< 0.25$ %–1 % of the methane production. These areas, in general, are characterized by significant unconventional oil and gas production; examples are found in the Appalachian Basin (northeastern Pennsylvania and the tri-state corner of southern Pennsylvania, eastern Ohio, and northern West Virginia) as well as in the Permian Delaware and Midland sub-basins and in parts of the Haynesville, Eagle Ford, and Bakken (Fig. 8). We also estimate areas with excessive methane loss rates ($> 10$ % of methane production; Fig. 8) in each major producing basin, particularly in the Appalachian Basin, in the Michigan Basin, and in the Greater Anadarko area of Missouri (Fig. 8). High methane loss rates are likely linked to the predominance of old, leak-prone, low-producing well sites (e.g., in parts of the Appalachian and

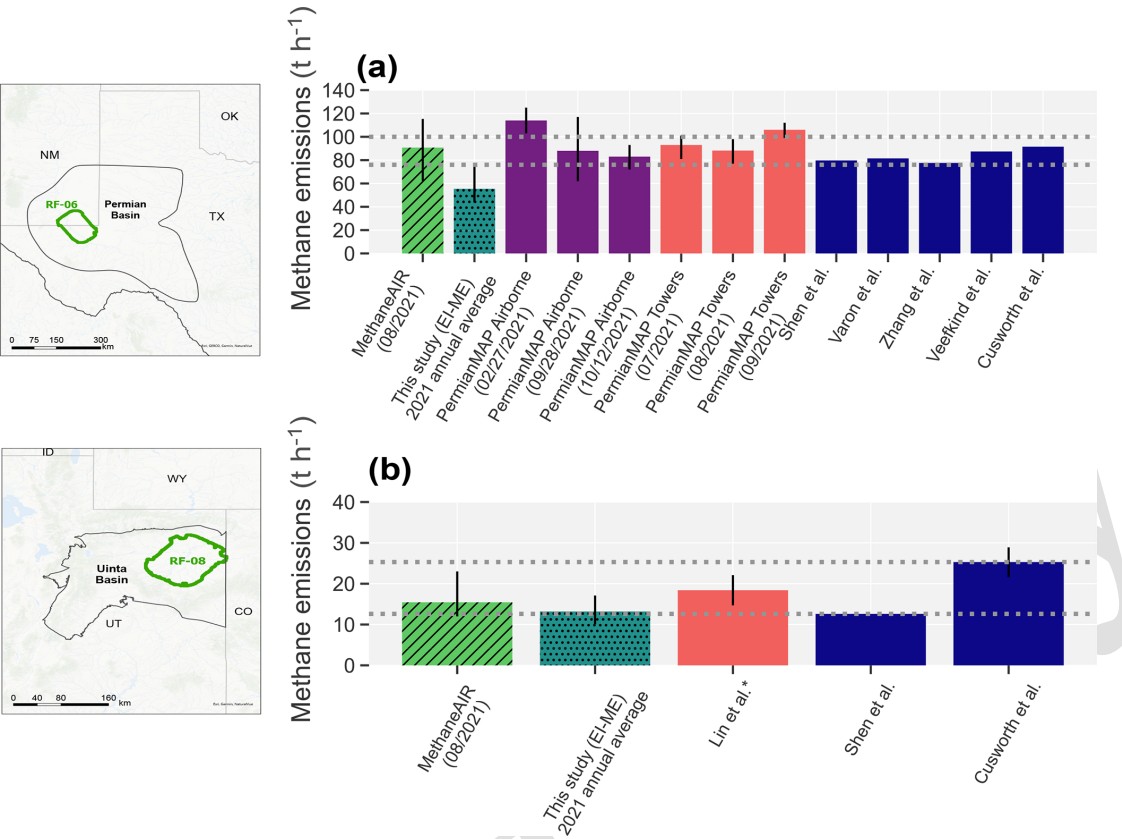

**Figure 6.** Comparison of the EI-ME inventory with MethaneAIR and other peer-reviewed studies for two sub-regions of the Permian and Uinta basins. Bars are color coded by emissions quantification method (MethaneAIR – hatched green bar; EI-ME – hatched dark green; PermianMAP airborne studies – purple; PermianMAP or tower-based study – red color; TROPOMI studies – dark blue bars). Lin et al. (2021) report total Uinta Basin methane emission estimates; we adjusted their estimate to account for the ratio of gas production in the RF-08 region to the total gas produced in Uintah and Duchesne counties in 2021 (RF-08 accounts for 74 % of the total production in the two counties). For all other studies, we use only the reported emission estimates that overlap with the MethaneAIR target boundaries. The dashed horizontal lines show a representative range of sub-basin methane emissions, computed via a bootstrapping procedure from all previously reported methane emissions (including EI-ME results) to derive a lower bound and an upper bound on the mean total methane emissions based on the 2.5th and 97.5th percentiles of the resulting bootstrap distribution, respectively. Map credit: ESRI, 2023.

San Joaquin basins – although the overall lack of correlation between absolute methane emissions [kg h$^{-1}$] and CE6 site age in Fig. S8 should be noted; also see Brantley et al., 2014) or may be associated with modeled midstream infrastructure emissions from sources not collocated with significant oil and gas production.

The updated 2020 gridded EPA GHGI inventory for oil and gas systems (Maasakkers et al., 2023) uses the same source of oil and gas activity data as this study (Enverus, 2024) and allocates GHGI emissions to specific emission source categories using infrastructure locations and methane emission scaling factors (e.g., scaled using well count and oil and/or gas production for well sites, depending on the source category). The estimated methane emission hotspots (Fig. 7b) are in reasonable agreement with this study's estimated spatial distributions ($r = 0.64$; Fig. 7a), with notable exceptions in parts of the Michigan Basin (Michigan),

the Appalachian Basin (Pennsylvania, Ohio, and West Virginia), the Powder River Basin (Wyoming), and the Barnett (east Texas), the Permian (west Texas), and the San Joaquin (southern California) basins (Fig. 7b). In parts of these basins, strong methane hotspots appear in regions that likely reflect a dependence of emissions spatial allocation on the spatial density of infrastructure (Fig. S5). This differs from this study's spatial allocation, which not only leverages infrastructure locations but simultaneously integrates the empirically observed facility-level methane emission characteristics (Figs. 1, 2), which can vary among populations in the same facility category (e.g., the distinction between emission profiles for low/non-low production well sites or among different production cohorts of the non-low production well site category).

In general, we find large differences in the magnitude of methane emissions in all of the major basins shown in

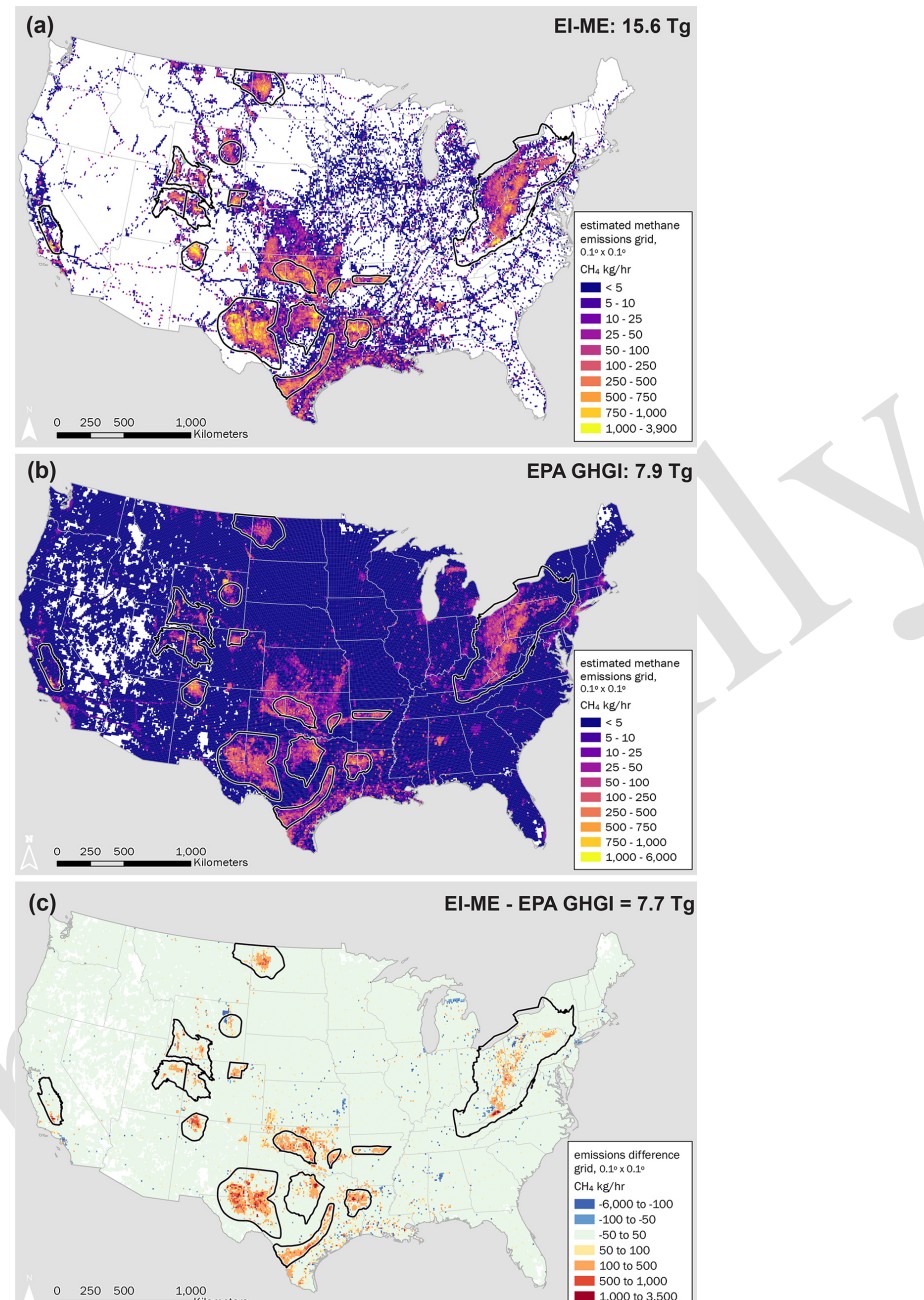

**Figure 7.** Estimated spatial distribution of total methane emissions and comparison with the EPA GHGI estimates. **(a)** This study's assessment of the spatial distribution of US total oil and gas methane emissions, showing the estimates for the contiguous US (excluding Alaska, the total estimated methane emissions were 15.6 Tg in 2021). For visualization and comparison with the EPA GHGI inventory, the total methane emissions are gridded to a $0.1° \times 0.1°$ spatial scale ($\sim 10\,\mathrm{km} \times 10\,\mathrm{km}$). Major basin boundaries are outlined using black polygons. **(b)** Estimated spatial distribution of total oil and gas methane emissions based on the EPA GHGI (2020; Maasakkers et al., 2023). Note that the EPA GHGI data shown here are for the year 2020, the latest year for which spatially explicit data are available. **(c)** Difference in spatially explicit methane emissions between this study's estimates and the EPA GHGI. Warmer colors indicate comparatively high estimates from this study relative to the EPA GHGI. We acknowledge that the comparison is limited by the different time periods of the two studies – 2021 in this study versus 2020 for the EPA GHGI. Nevertheless, as both studies report annual averages, it is unlikely that significant differences in aggregate spatial distribution would have occurred between 2020 and 2021 that would alter the main conclusions from this analysis. For the EI-ME, uncertainty estimates for each grid cell (i.e., lower bound and upper bound on mean estimates) are presented in map form in Fig. S9. Map credit: ESRI, 2023. Basin boundaries are based on US EIA basin boundaries data (https://www.eia.gov/maps/maps.php, last access: 22 August 2024).

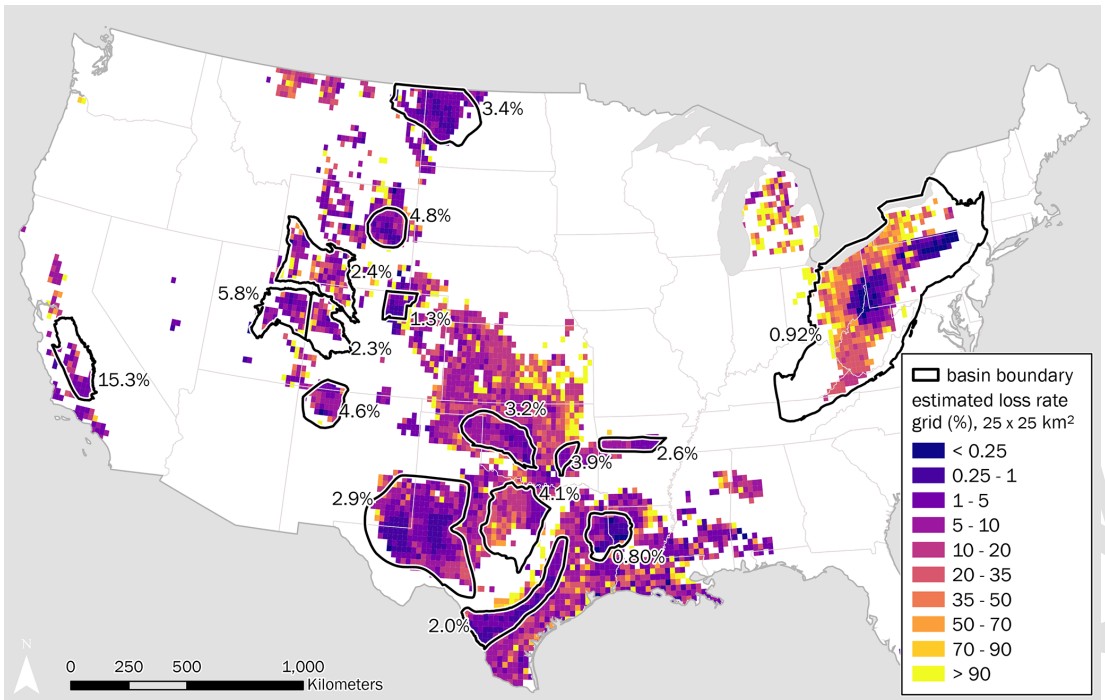

**Figure 8.** Estimated mean spatial distribution of production-normalized methane loss rates. For ease of visualization, we aggregate our facility-level methane inventories to a coarser spatial grid ($25 \times 25\ \text{km}^2$) and normalize each grid's total estimated methane emissions relative to total methane production to derive spatially explicit methane loss rates, assuming an 80 % methane content in natural gas. Major basin boundaries are outlined in black, and mean basin-level methane loss rates are shown as %. Map credit: ESRI, 2023. Basin boundaries are based on US EIA basin boundaries data (https://www.eia.gov/maps/maps.php, last access: 22 August 2024).

Fig. 7 when comparing the spatially explicit methane emissions in the GHGI and this study's estimates (Figs. 7c, S6). We also find large differences in the spatial distribution of the methane emissions when comparing this study's spatially explicit emissions inventory with the EDGAR v8 inventory (EDGAR, 2023; Fig. S7). Note that the EDGAR v8 total methane emissions are similar in magnitude to the EPA GHGI inventory estimates (Fig. 1), although emissions spatial allocation methods are primarily dependent on scaling by oil production characteristics, such that large methane hotspots are estimated to be located in the oil-dominant basins of the Permian, the Bakken, and the Eagle Ford (Fig. S7).

Our results suggest both an underestimation of the magnitude of spatially explicit emissions in key US oil and gas basins as well as potentially unrepresentative spatial distributions of these emissions in the EDGAR v8 and the EPA GHGI gridded inventories. These results carry important implications for the use of traditional bottom-up inventories as a priori information in Bayesian inversions of satellite observations for methane quantification since both the magnitude and the spatial allocation of emissions could influence the posterior results from these modeling systems under certain observational data constraints, such as insufficient observational data density (Shen et al., 2022).

## 4  Data availability

EI-ME_v1.0 can be accessed at https://doi.org/10.5281/zenodo.10734299 (Omara, 2024) in the open-access GeoPackage file format. The GeoPackage file includes estimates for Alaska, while a .netdf file is also provided with gridded emission results for only the lower 48 US states to facilitate easier comparison with recent satellite-derived methane emission estimates for the contiguous US (Shen et al., 2022; Lu et al., 2023).

## 5  Code availability

The Python 3.7 code used for emissions modeling, extrapolation to populations of facilities, and data visualization is available from the corresponding author upon reasonable request.

## 6   Conclusions

Accurate and comprehensive assessment of oil and gas methane emissions is pivotal in informing effective methane mitigation policies. In this study, we develop robust statistical models based on measured facility-level methane emissions and integrate these models with comprehensive oil and gas activity data for onshore US oil and gas facilities to estimate total national oil and gas methane emissions for the year 2021. We estimate a total of $\sim$ 16 (14–18) Tg of oil and gas methane emissions in 2021, representing a mean methane loss rate of 2.6 % of gross gas production. Our national methane emission estimate, while in reasonable agreement with previous measurement-based estimates using facility-level measurements and satellite observations, is nevertheless roughly 2× greater than official inventories from the EPA Greenhouse Gas Inventory (GHGI). This improved assessment of national methane emissions underscores the importance of integrating measurement-based data to develop robust methane emissions inventories which, as we show in this work, exhibit substantial variability in both the magnitude and spatial distribution of total methane emissions across major oil and gas basins.

Further improvements to methane emissions inventories are possible through greater integration of measurement-based data, including remote-sensing approaches that can provide comprehensive area-wide total methane emissions, quantifications of high-emitting methane point sources, and high-resolution spatial disaggregation of total methane emissions. In this study, we present the first set of such remote-sensing quantifications, based on MethaneAIR measurements in sub-basins of the Permian and Uinta, and we demonstrate reasonable agreement with several previous peer-reviewed assessments of total area methane emissions over similar spatial domains and time periods. These comprehensive area-wide assessments also enable a detailed characterization of the importance of small methane sources dispersed across regions viz-à-viz large, concentrated methane point sources, revealing their relative importance and variability across unique US oil- and gas-producing basins.

The EI-ME inventory provides a detailed characterization of total methane emissions by key facility category at the national level as well as at the regional/basin level, thus helping provide policy-relevant information that is important in developing and tracking effective methane mitigation strategies. The quantified uncertainties in our methane emission estimates could be improved upon in future studies through additional peer-reviewed data collection efforts, which are needed to develop further insights in response to ongoing methane mitigation efforts. There is a research need to develop robust statistical methods for the effective integration of lower-detection-limit ground-based facility-level methane emissions data (such as the data synthesized herein) with the growing number of airborne facility-level measurement studies, which generally have higher method detection limits (e.g., the airborne methane remote-sensing data in Duren et al., 2019; Cusworth et al., 2022; Sherwin et al., 2024). As demonstrated herein, improved integrated assessments of facility-level, regional, and national methane emissions inventories, based on measurement data, support ongoing efforts to accurately quantify methane emissions, identify key methane sources and regions for targeted methane reductions, and track progress toward methane reduction goals.

**Supplement.** The supplement related to this article is available online at: https://doi.org/10.5194/essd-16-1-2024-supplement.

**Author contributions.** MO and RG conceptualized the study. MO developed the facility-level methane emissions models. Formal data analysis, interpretation, and visualization were performed by MO, AH, and RG, with assistance from KM and JPW. MethaneAIR GIM modeling and high-emitting methane point-source assessments were performed by JB, MS, and SCW. MO wrote the manuscript with contributions from all co-authors.

**Competing interests.** The contact author has declared that none of the authors has any competing interests.

**Disclaimer.** Publisher's note: Copernicus Publications remains neutral with regard to jurisdictional claims made in the text, published maps, institutional affiliations, or any other geographical representation in this paper. While Copernicus Publications makes every effort to include appropriate place names, the final responsibility lies with the authors.

**Acknowledgements.** We thank Madeleine A. O'Brien for providing the OGIM database and related geospatial-data gridding script. We are grateful to Daniel Zavala-Araiza for providing comments.

**Financial support.** This work was made possible by support from the Bezos Earth Fund.

**Review statement.** This paper was edited by Jing Wei and reviewed by three anonymous referees.

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

## Remarks from the language copy-editor