# Peer review of "Constructing a measurement-based spatially explicit inventory of US oil and gas methane emissions (2021)"

_Earth System Science Data, 2024_

## Author Comment (AC1)

**Author response to reviewer comments**

**Anonymous Referee # 3**

The authors developed a detailed compilation of activity data and emissions measurements to estimate 2021 US oil and gas methane emissions. The measurements were drawn from published studies and used to develop emission rate distributions. These distributions were used to estimate methane emissions at the facility level and then, estimate methane emissions at the national level. The authors also compare their results to airborne measurements using MethaneAIR. Overall, their spatially explicitly inventory is a valuable contribution, especially for remote measurements using satellites. In addition, they find important spatial trends that can be used to improve emission estimates and inform mitigation. Below are some high-level and detailed comments that can improve clarity of the paper.

We thank Reviewer #3 for these detailed comments and review of our manuscript. We provide below point-by-point responses.

High-level comments:

The authors use measurement data from various years (not 2021) to estimate methane emissions for 2021. However, due to regulations, technology advancements, and other factors, methane emissions distributions may be changing over time, as the authors acknowledge. It doesn't appear that the authors try to correct for this temporal variability or address this in their discussion.

As we acknowledged in the Main Text, specific uncertainties related to the impact of changing operator practices and/or promulgated regulations of oil and gas methane emissions are difficult to quantify due to lack of data. However, for well sites (the largest contributor to the estimated methane emissions in this study), the large body of evidence from the measurement data that we synthesized, which span the years post-2011 EPA NSPS to 2020, do suggest that (i) for low production well sites, absolute methane emissions are weakly correlated with production rates and (ii) for non-low production well sites, newly developed high producing well sites exhibit lower methane loss rates (absolute emissions normalized by production) compared with aging and lower producing sites in this category. We leverage these insights based on empirical observations, which we assume generally hold true across basins, to model national methane emission rates, with specific application to activity data (both well site count and production rates) in 2021. As such, temporal variability that are predictable as part of the insights derived from these empirical observations (i.e., how do facility-level emissions sizes vary with changes in production rates—e.g., Figure 1a) are implicitly accounted for by constraining estimates to activity data specific to 2021.

We have included the following sentences in Section 3.2 to clarify:

> "As noted previously, basin-level differences in total methane emissions could also be impacted by federal/state-level regulations of oil and gas methane emissions and/or operator-specific practices, affecting both the magnitude and temporal variability in emissions. While our methods are based on insights derived from empirical observations and robust modelling to estimate methane

*emissions specific to oil and gas activity in 2021, we lack sufficient data to characterize the impacts of specific regulations or operator practices. Further studies are needed to assess oil and gas methane emission trends and corresponding underlying drivers."*

We provide further details below in response to similar comments from Reviewer #3.

The definition of the methane loss rate is unclear. Although the authors mention in the Results (though I would have expected this in the Methods) that an 80% methane content was assumed, it remains unclear how the conversion was done for oil facilities.

We include the following clarification text in Section 2.4 of Methods:

*"We compute basin-level and national methane loss rates as the ratio of estimated basin-level methane emissions to gross methane production in 2021, based on gross natural gas production*
*data from Enverus Prism (Enverus, 2024) and an assumed average methane content of 80% in natural gas. Our assumption of an average 80% methane content in natural gas is informed by regional estimates of methane composition in natural gas based on the EPA GHGI (EPA, 2022). We acknowledge that uncertainties in methane composition across basins likely increases uncertainties in our overall methane loss rate calculations. Further studies on basin-level methane*
*composition are needed to constrain these uncertainties. This methane intensity metric allows for a direct comparison of estimated methane losses relative to gross methane production across different basins. While our use of gross methane production accounts for emissions from associated gas produced during oil operations, the results are not intended to represent lifecycle emission intensities, which are outside the scope of this work."*

The authors need to provide a clear definition for "measurement-based inventories". The authors do not measure all sources but use methane emission rate distributions based on available measurements, which does not cover all sites but some subset. Therefore, the question is how many and which measurements are needed to have a representative sample that can be used to create "measurement-based inventories".

We appreciate Reviewer # 3's comment on clarifying our definition for "measurement-based inventories." We include the following revisions in the Introduction Section 1:

*"In this work, we utilize previous peer-reviewed facility-level measurement data for methane*
*emissions at oil and gas facilities in the major US oil and gas production basins to develop an improved assessment of national, basin-level, and facility-level methane emissions based on oil and gas activity in 2021.* *Our measurement-based inventory differs from other "bottom-up" inventories that use generic emission factors (e.g., EPA GHGI) in that we leverage empirical observations to derive insights on facility-level methane emission distributions useful for estimating population*
*mean total methane emissions."*

Previous research have used a limited sample of measurement-based data on facility-level oil and gas methane emissions to develop an improved inventory of basin-level or regional methane emissions. For example, Zavala-Araiza et al. (2016) used facility-level measurement datasets of the order of hundreds of measurements in combination with a robust statistical model to characterize facility-level methane emission distributions and estimate the total Barnett oil and gas methane emissions, which were then validated (and showed good agreement) with independent airborne measurements. Alvarez et al. (2018) used facility-level methane measurements from 433 production sites and measurements at other facility types (e.g., compressor stations, processing plants) to develop emission models to estimate basin-level methane emissions that were validated with independent airborne measurements, and the insights from these models were used to estimate national methane emissions conditional on oil and gas activity data in 2015. These works (Zavala-Araiza et al., 2015 and Alvarez et al., 2018) demonstrated that measurement-based inventories developed using these methods produce results that are in good agreement, within statistical uncertainty, of independent airborne measurements of total area methane emissions. Our study follows similar approaches and yields results that are comparable, within statistical uncertainty, of independent airborne and satellite-based estimates, as we discuss in the Results and Discussion section.

To characterize how many samples are required to minimize uncertainties in the development of regional measurement-based inventories, well-designed coordinated measurement campaigns, employing multiscale measurements from ground-based facility-level measurements to top-down airborne/satellite measurements would be needed. Such assessment and related analyses are beyond the scope of this work.

We have revised Section 2.3 of our manuscript to include the following clarification texts and discussion of sample representativeness:

*"Our approach for estimating regional and national oil and gas methane emissions builds upon previous works that used data from hundreds to thousands of ground-based facility-level measurements (Zavala-Araiza et al., 2015; Alvarez et al., 2018; Omara et al. 2018; Omara et al., 2022) in combination with robust probabilistic models integrated with oil and gas activity data. Zavala-Araiza et al. (2015) and Alvarez et al. (2018) demonstrated that measurement-based inventories developed using these methods produce total methane emission results that are in good agreement, within statistical uncertainty, of independent airborne measurements of total area methane emissions.*

*For non-low production well sites (average facility-level production rates > 15 boed), we begin by evaluating facility representativeness on the basis of (i) geographical diversity of measurements, (ii) distribution of facility-level production rates of measurements compared with the national population of well site facilities, and (iii) the distribution of facility-level methane emission rates across basins (Supplemental Fig. 3). Our measurement data, while limited in sample size, covers eight major US oil and gas basins with diverse oil and gas production characteristics, including the Appalachian, the Permian, Uinta, Barnett, Fayetteville, Greater Green River, and Denver-Julesburg. The wide range in basin-level gas-to-oil ratios (~1 to 800 Mcf/barrel) is well represented in the data for the sampled basins (Supplementary Fig. 3b).*

*In addition, the distribution of facility-level natural gas production rates shows reasonable overlap with that for the national population of non-low production facilities, and the broad range in distribution of facility-level production rates across the national population of sites (~90 Mcfd to >50,000 Mcfd) is well represented in the sampled sites (Supplementary Fig. 3c). However, the distribution of production rates for the sampled sites suggests potential bias toward higher-producing sites relative to the national distribution (Supplementary Fig. 3c). We account for any such potential biases by developing emission models based on production-normalized methane loss*

*rate distributions (methane emitted relative to methane produced) across seven cohorts of specific*

*gas production rates (further details below).*

*We develop and use probabilistic emission rate distributions based on production-normalized methane loss rates, which shows a wide range <0.01% to >90% (Figure 1a) across all basins (Supplementary Fig. 3d), reflecting, in part, the diversity in production characteristics within and across basins. We use production-normalized methane loss rate distributions because (i) the*

*empirical data across a wide diversity of oil and gas production facilities suggests an inverse relationship in which high-producing facilities exhibit comparatively lower methane loss rates, and vice versa (Figure 1a) and (ii) the consolidated dataset includes measurements collected in earlier years before 2021. By using the production-normalized methane loss rate distribution models for specific cohorts of facility-level production rates, we do not model any particular site that is active*

*in 2021 as exhibiting the same emission rate size as observed when measurements were taken in the past, as the empirical data and the model constrains facility-level methane loss rates to production levels, which will be time-variant. As such, we provide a necessary constraint on our estimates, effectively adjusting modelled facility-level methane emission rates if production rates have substantially changed over time."*

[Figure]

**Supplementary Fig. 3:** Geographical coverage, distribution of facility-level production rates, and emission rates for well site measurements used in this study.

**a.** The map shows the well site oil and gas production data for 2021, color-coded by combined oil and gas production rates (boe/year). Major basins for which substantial measurement-based data on oil and gas methane emissions are available are highlighted in red. **b.** Assessment of the basin-level production characteristics, based on average gas-to-oil ratios in Mcf/barrel in 2021. The bar plots show the basin-level GOR ratios, light pink bars correspond to basins for which measurement-based data (see Main Text on criteria) are available. The number of samples are shown in top x axis in red. The solid blue line shows the average GOR ratio for all sites in the US in 2021 (average of 11 Mcf/bbl) and the dotted dark red lines show the minimum (5 Mcf/barrel) and maximum (230 Mcf/barrel) for all the basins for which we have measurement-based data. The right y-axis shows the % of total US onshore BOE production that is accounted for by each major US basin. **c.** Histogram of gas production rates comparing the distribution for the sampled non-low production sites with that for the population of non-low production sites in the US. **d.** Comparison of empirical distribution of facility-level production-normalized methane loss rates for the major basins for which measurements are available. We performed non-parametric bootstrap resampling, with replacement, of the data for each basin, repeated $10^4$ times to generate the likely extent or uncertainties of the distributions conditional on empirical observations. For each basin, we plot the distributions based on the $10^4$ bootstrap results.

Detailed Comments:

L16: Provide a clear definition of "measurement-based inventories". It will be helpful to clarify the extent to which measurements are conducted for the inventory to be considered "measurement-based".

Please see author response above.

L18: How is "representativeness" assessed?

Please see author response above.

L20: How is the comprehensiveness of the spatial data assessed?

Comprehensiveness of spatial data is based on the comparison of our spatially explicit data (sourced primarily from Enverus Prism, a proprietary data source, www.enverus.com) with reported metrics in official energy statistics. For example, the US EIA reports a total gross national gas production of ~42 Tcf in 2021 (https://www.eia.gov/dnav/ng/ng_prod_sum_dc_NUS_mmcf_a.htm) consistent with the totals from the spatially explicit database (wellheads with allocated production data) available from Enverus Prism (~42 Tcf in 2021). Similarly, our spatially-explicit data for midstream facilities are generally consistent with official estimates from the EPA GHGI (for example, the GHGI reports a total of ~2,000 transmission compressor stations, similar to the spatially explicit activity data used in this work). In Section 2.5 of the Main Text, we acknowledge that there could be uncertainties in oil and gas activity data that are difficult to quantify because much of this information is based on operator-reported data, with unknown uncertainties.

We include the following clarification sentences in the revised manuscript in Section 2.1:

> *"We consider these spatial data as comprehensive for the US oil and gas production well sites as it is consistent with the official gross oil and gas production reported by the US Energy Information Administration for 2021 (e.g., the sum of gross gas production from spatially explicit well-level production data from Enverus Prism is consistent with the total of ~42 Tcf of US natural gas gross withdrawals reported by the US Energy Information Administration, https://www.eia.gov/dnav/ng/ng_prod_sum_dc_NUS_mmcf_a.htm)."*

L25: Is production data at the annual level?

Yes. We have revised this sentence for clarity:

> *"We then integrate these emissions data with comprehensive spatial data on national oil and gas activity to estimate each facility's mean total methane emissions and uncertainties for the year 2021, from which we develop a mean estimate of annual national methane emissions, resolved at 0.1° × 0.1° spatial scales (~10 km × 10 km)."*

L28-32: Very long sentence. Break sentence in two.

We have revised this sentence as follows:

> *"Additionally, we present and compare novel comprehensive wide-area airborne remote sensing*
> *data and results of total area methane emissions and the relative contributions of diffuse and*
*concentrated methane point sources as quantified using MethaneAIR in 2021. The MethaneAIR*
> *assessment showed reasonable agreement with independent regional methane quantification*
> *results in sub-regions of the Permian and Uinta basins and indicated that diffuse area sources*
> *accounted for the majority of total oil and gas emissions in these two regions."*

L46: Countries submit national inventory reports to the UNFCCC but there is no UNFCCC Greenhouse
Gas Inventory. They are required to follow IPCC guidelines.

We have revised this sentence as follows:

> *"At the national level, methane inventories are typically developed using "bottom-up" methods,*
*for example, these methods are used by most countries that report annual inventories to the*
> *UNFCCC (UNFCCC, 2023)."*

  L53: Define what is meant by "measurement-based inventories". See previous and high-level comments.

Please see response above. We have included the following sentence for clarification:

> *"In this work, we utilize previous peer-reviewed facility-level measurement data for methane*
> *emissions at oil and gas facilities in the major US oil and gas production basins to develop an*
> *improved assessment of national, basin-level, and facility-level methane emissions based on oil and*
*gas activity in 2021. Our measurement-based inventory differs from other "bottom-up" inventories*
> *that use generic emission factors (e.g., EPA GHGI) in that we leverage empirical observations to*
> *derive insights on facility-level methane emission distributions useful for estimating population*
> *mean total methane emissions."*

L64: Reading on, it appears that the paper uses measurements not from 2021. How is the data corrected for
temporal variability?

Temporal variability in facility-level methane emission rates could be influenced by several factors,
including changes in facility operations (e.g., increased or decreased production activities over time,
installation of emission control devices, or removal of auxiliary processing equipment, etc), frequency of
scheduled or unscheduled maintenance activities leading to intentionally vented emissions, regulatory
requirements on emission controls, voluntary efforts on emission controls, etc. We unfortunately lack the
data needed to assess facility-level methane emission trends. However, for well sites—which accounts for
the majority of our estimated emissions—the one important attribute that is consistently reported across the
consolidated measurements in our study is the facility-level production rate. It is important to note that
production declines substantially over time at a well site over time, and our consolidation of facility-level
emission rate data for a wide range of production rates allows us to develop key insights that make it possible to use data collected in previous years to estimate emissions for the year 2021. Specifically, for the non-low production well site category, by using stratified methane loss rate distributions for specific cohorts of production rates (see Methods), we do not model any particular site that is active in 2021 as exhibiting the exact emission rate size exactly as observed when measurements were taken in the past, as the empirical data and the model constrains facility-level methane loss rates to production levels, which will be time-variant. This provides a necessary constraint on our model regarding temporal variability, but as we have acknowledged, our results will have uncertainties that are difficult to quantify and related to, among other factors, potential changes in emissions as regulations are enacted and implemented. Further comprehensive studies on total area methane emissions are needed to better understand temporal variability in emissions.

*"We develop and use probabilistic emission rate distributions based on production-normalized methane loss rates, which shows a wide range <0.01% to >90% (Figure 1a) across all basins (Supplementary Fig. 3d), reflecting, in part, the diversity in production characteristics within and across basins. We use production-normalized methane loss rate distributions because (i) the empirical data across a wide diversity of oil and gas production facilities suggests an inverse relationship in which high-producing facilities exhibit comparatively lower methane loss rates, and vice versa (Figure 1a) and (ii) the consolidated dataset includes measurements collected in earlier years before 2021. By using the production-normalized methane loss rate distribution mode for specific cohorts of facility-level production rates, we do not model any particular site that is active in 2021 as exhibiting the same emission rate size as observed when measurements were taken in the past, as the empirical data and the model constrains facility-level methane loss rates to production levels, which will be time-variant. As such, we provide a necessary constraint on our estimates, effectively adjusting modelled facility-level methane emission rates if production rates have substantially changed over time."*

L65: Measurement data from which year(s)?

We have revised this sentence as follows:
*"First, we develop statistically robust facility-level methane emission models based on measurement data collected in the years post-2011 (when EPA's NSPS were first proposed) through 2020. We use these models to estimate national methane emissions, on both an absolute basis (Tg/year) and production-normalized basis (% emitted relative to methane production)."*

L67: relative to methane or natural gas production?

Production-normalized methane loss rates are relative to methane production.

L81-83: Above, the authors mention that the loss rates are normalized by methane production. How is oil/gas production converted to methane production? Is the production data for 2021 used or is the production data corresponding to the month/year of measurement used?

For the measurement-based data, methane loss rates, as reported in the different studies, are based on methane production specific to the time in which measurements occurred (averaged to hourly production rates based on the production data for the month in which measurements occurred because production data are generally reported on a monthly basis).

L78-91 describes the assessment of the production characteristics (annual production for 2021, which is then expressed as a daily average in Mcfd, bbl per day, etc based on the number of production days in the year). This assessment is specific to the national population of well sites, based on monthly data that is reported at the well-level.

L141 describes the method for calculating methane loss rate, which is defined methane emitted relative to methane produced. In this study, methane produced at a well site facility is a factor of gross gas production and the methane content. Further details can be found in the revised Section 2.4, which includes the following paragraph on computation of basin-level methane loss rates:

*"We compute basin-level and national methane loss rates as the ratio of estimated basin-level methane emissions to gross methane production in 2021, based on gross natural gas production data from Enverus Prism (Enverus, 2024) and an assumed average methane content of 80% in natural gas. Our assumption of an average 80% methane content in natural gas is informed by regional estimates of methane composition in natural gas based on the EPA GHGI (EPA, 2022).*
*We acknowledge that uncertainties in methane composition across basins likely increases uncertainties in our overall methane loss rate calculations. Further studies on basin-level methane composition are needed to constrain these uncertainties. This methane intensity metric allows for a direct comparison of estimated methane losses relative to gross methane production across different basins. While our use of gross methane production accounts for emissions from associated*
*gas produced during oil operations, the results are not intended to represent lifecycle emission intensities, which are outside the scope of this work."*

Table 1: Are the estimated total methane emissions reported in column 6 done by the authors here in this
paper or are these previous results? If they are estimated in this paper, they are better placed in the results.

The EI-ME estimated total methane emissions in Table 1 are new results estimated as part of this study. They are included in this Table at this point in the manuscript to contextualize both the activity data for various infrastructure categories, the methane measurement data sources, as well as the EPA's GHGI
estimates. The discussion of these results is presented in the Results and Discussion section, supplemented with relevant figures.

L130-135: Many of these measurements were conducted before 2021. There needs to be a description as to how these measurements can be used to estimate emissions in 2021, and if some adjustments are needed.

Please see detailed responses above.

L141: what are the units for the methane loss rate? If it's unitless, it should say so. Is "CH4" methane lost or measured? What is "Gas"? All the variables here need to defined and their units clearly provided after
the equation.

*The production-normalized methane loss rate is unitless. We have revised the equation as follows:*

$$methane\ loss\ rate\ [unitless] = CH_4[\frac{kg}{h}] \times \frac{1}{Gas\ [Mcfd]} \times \frac{1\ Mcf}{19.2\ [kg\ CH_4]} \times \frac{1}{\sigma_{CH4}} \times \frac{24h}{1d}$$

*where $CH_4$ is the measured facility-level methane emission rate, Gas [Mcfd] is the reported gas production rate in thousand cubic feet [Mcf] per day, 19.2 kg/Mcf is the methane density at 60 °F (15.5 °C) and 1 atm, and $\sigma_{CH4}$ is the assumed methane fraction in the produced natural gas (we assume an average of 80% methane content in the produced natural gas).*

Figure 2. These distributions are better placed in the Results section.

We view these distributions as an important component of the methods and models for estimating total
methane emissions. We focus the Results and Discussion section on the outputs of these models.

Figure 3. The K-S test provides a measure of goodness of fit. How can it be used to assess the representativeness of the underly methane emissions measurements?

The inset figure showing distribution and K-S test results was not intended to suggest a method for assessing facility representativeness. The goodness of fit tests occurs as part of the emission distribution modeling. We have revised this general schematic as follows:

[Figure]

Figure 3. There are some arrows missing. I suggest an arrow be added to the black line from the dashed rounded box to the site-level CH4 emission rate histogram.

We thank Reviewer 3 for this suggestion. We have revised Figure 3 accordingly.

L240: How was 500 selected?

was selected as a reasonably simulation size that is not too computationally intensive to implement but that also gives sufficient statistical power to develop robust model uncertainty assessment.

We have included the following clarification sentence:

*"We use 500 simulation results for each facility as a reasonable simulation size that is not too computationally intensive to implement but that also gives sufficient statistical power to develop robust model uncertainty assessment."*

L260-261: How was data limitation determined? There are published studies on downstream natural gas, post-meter, and abandoned well emissions. Are the authors looking for some specific number of measurements?

Given the focus of our study on developing spatially-explicit measurement-based methane emissions inventory, we did not include these sources due to a general lack of comprehensive spatially explicit activity data.

We have revised the sentence to clarify the lack of comprehensive spatially explicit data for these sources:

*"In addition, due to lack of comprehensive spatially explicit data, our measurement-based inventory does not include methane emissions from downstream natural gas distribution, LNG storage, post-meter emissions, and abandoned oil and gas wells."*

L271: Remove the word "However"

We have revised L271 as follows:

*"In addition, consistent with previous findings (Alvarez et al., 2018; Rutherford et al., 2021; Shen et al., 2022), our central estimate is significantly greater than inventories developed using the traditional bottom-up source-level emission factor approaches: we find a factor of 1.9× and 1.8× greater total methane emissions than is estimated by the EPA Greenhouse Gas Inventory (EPA, 2022) and EDGAR v8 (EDGAR, 2023) inventories for the year 2021. (Fig. 5a)."*

L288-289: methane content in natural gas can be variable. Does the 95% CI include methane content variability or is it assume to be fixed at 80%?

Nationally and across regions we assume an average 80% methane content in natural gas.

We include the following paragraph in Section 2.4 to clarify:

*"We compute basin-level and national methane loss rates as the ratio of estimated basin-level methane emissions to gross methane production in 2021, based on gross natural gas production data from Enverus Prism (Enverus, 2024) and an assumed average methane content of 80% in*
*natural gas. Our assumption of an average 80% methane content in natural gas is informed by regional estimates of methane composition in natural gas based on the EPA GHGI (EPA, 2022). We acknowledge that uncertainties in methane composition across basins likely increases uncertainties in our overall methane loss rate calculations. Further studies on basin-level methane composition are needed to constrain these uncertainties. This methane intensity metric allows for*
*a direct comparison of estimated methane losses relative to gross methane production across different basins. While our use of gross methane production accounts for emissions from associated gas produced during oil operations, the results are not intended to represent lifecycle emission intensities, which are outside the scope of this work."*

Fig. 4: The legends should be moved outside of the plot. It would be helpful if the groupings of bars separated by dashed lines were annotated – e.g., green bars should just be labeled GOSAT.

We have updated Fig. 4 as follows:

[Figure]

**Figure 4.** Comparison of this study's national estimate of total methane emissions from the oil and gas supply chain with previous measurement-based estimates. The first three bars show the oil and gas methane emissions estimated from facility-level measurements (this study, Alvarez et al. 2018) and production-sector-only methane emissions estimate by Rutherford et al. (2021) using models developed from component-level measurement data. Blue bars show the estimated emissions for the production sector, gold bars show the estimated emissions for the midstream and downstream facilities (compressor stations, processing plants, refineries, gathering and transmission pipelines). Error bars show the estimated 95% confidence bounds on the mean total methane emissions estimates. This study's estimate of total national methane emissions include ~0.1 Tg/year of estimated methane emissions for Alaska. The green bars and the red bars show the satellite-derived estimates for contiguous US based on GOSAT and TROPOMI observations, respectively. The last two bars show the "bottom-up" inventories from EPA GHGI and EDGAR v8 for the contiguous US. In all cases, the years for which methane emissions are estimated are shown on the top x-axis.

L294-299: The caption describes the first three bars only but should describe the rest as well.

We have revised Fig. 4 caption as follows:

> *"**Figure 4.** Comparison of this study's national estimate of total methane emissions from the oil
> and gas supply chain with previous measurement-based estimates. The first three bars show the oil
> 455     and gas methane emissions estimated from facility-level measurements (this study, Alvarez et al.
> 2018) and production-sector-only methane emissions estimate by Rutherford et al. (2021) using
> component-level measurement data. Blue bars show the estimated emissions for the production*

*sector, gold bars show the estimated emissions for the midstream and downstream facilities (compressor stations, processing plants, refineries, gathering and transmission pipelines). Error bars show the estimated 95% confidence bounds on the mean total methane emissions estimates. This study's estimate of total national methane emissions include ~0.1 Tg/year of estimated methane emissions for Alaska. The green bars and the red bars show the satellite-derived estimates for contiguous US based on GOSAT and TROPOMI observations, respectively. The last two bars show the "bottom-up" inventories from EPA GHGI and EDGAR v8 for the contiguous US."*

L345-346: If weakly correlated, should factors other than infrastructure be considered?

Other possible predictors of facility-level methane emissions could indeed be assessed; unfortunately, such data and related attributes were unavailable in the reported facility-level measurement data synthesized herein.

L501: How are production rates determined for midstream infrastructure?

Our estimate of facility-level emissions for compressor stations and processing plants are independent of throughput rates. In our spatial aggregation of methane loss rates, there will be cases where emissions in certain locations are dominated by midstream infrastructure in those locations that are handling oil and gas produced from well sites that are located in a different grid. This is possible given the grid resolution of ~25 km x 25 km. In these cases, the methane loss rates could be much higher than would be if expected if the production from the well site infrastructure were collocated with the gathering/transportation infrastructure within the chosen grid resolution of ~25 km x 25 km.

L502-503: If this study uses the data for 2021, would it not be different from the 2018 gridded EPA GHGI inventory data?

We expect minor differences in GHGI total methane emissions year-to-year. We have updated our assessment to use the spatially explicit GHGI data for the latest year for which these data are now available, i.e., 2020. We acknowledge we are not comparing the estimates for the same year although we do not expect the overall conclusions to change. Not that this only affects our comparison of the spatially explicit inventory from the EPA GHGI (based on Maasakkers et al., 2023) which extends only up to the year 2020. The official report from the EPA GHGI does include the total inventory estimates for the year 2021, which we compare in Table 1 and in Figure 4.

L509: Figure 6b shows methane emissions for a sub-region of the Uintah Basin, for which the agreement was good. Therefore, I don't think it's the correct figure to be pointing to here.

We have revised this to reference Figure 7.

**References**

Alvarez, R. A., Zavala-Araiza, D., Lyon, D. R., Allen, D. T., Barkley, Z. R., Brandt, A. R., Davis, K. J., Herndon, S. C., Jacob, D. J., Karion, A., Kort, E. A., Lamb, B. K., Lauvaux, T., Maasakkers, J. D., Marchese, A. J., Omara, M., Pacala, S. W., Peischl, J., Robinson, A. L., Shepson, P. B., Sweeney, C., Townsend-Small, A., Wofsy, S. C., Hamburg, S. P. Assessment of Methane Emissions from the U.S. Oil and Gas Supply Chain. Science, 361, 186–188, https://doi.org/10.1126/science.aar7204, 2018.

EDGAR (Emissions Database for Global Atmospheric Research) Community GHG Database, a collaboration between the European Commission, Joint Research Centre (JRC), the International Energy Agency (IEA), and comprising IEA-EDGAR CO2, EDGAR CH4, EDGAR N2O, EDGAR F-GASES version 8.0, European Commission, JRC (Datasets), https://edgar.jrc.ec.europa.eu/report_2023, 2023.

Enverus Prism, www.enverus.com, last accessed February 06, 2024

EPA: United States Environmental Protection Agency, Inventory of US Greenhouse Gas Emissions and Sinks, https://www.epa.gov/ghgemissions/inventory-us-greenhouse-gas-emissions-and-sinks (last access: 20 December 2023), 2022.

Maasakkers, J. D., McDuffie, E. E., Sulprizio, M. P., Chen, C., Schultz, M., Brunelle, L., Thrush, R., Steller, J., Sherry, C., Jacob, D. J., Jeong, S., Irving, B., Weitz, M. A Gridded Inventory of Annual 2012–2018 U.S. Anthropogenic Methane Emissions. Environ. Sci. Technol., 57 (43), 16276–16288, https://doi.org/10.1021/acs.est.3c05138, 2023.

Shen, L., Gautam, R., Omara, M., Zavala-Araiza, D., Maasakkers, J. D., Scarpelli, T. R., Lorente, A., Lyon, D., Sheng, J., Varon, D. J., Nesser, H., Qu, Z., Lu, X., Sulprizio, M. P., Hamburg, S. P., Jacob, D. J. Satellite Quantification of Oil and Natural Gas Methane Emissions in the US and Canada Including Contributions from Individual Basins. Atmospheric Chem. Phys., 22 (17), 11203–11215, https://doi.org/10.5194/acp-22-11203-2022, 2022.

Zavala-Araiza, D.; Lyon, D. R.; Alvarez, R. A.; Davis, K. J.; Harriss, R.; Herndon, S. C.; Karion, A.; Kort, E. A.; Lamb, B. K.; Lan, X.; Marchese, A. J.; Pacala, S. W.; Robinson, A. L.; Shepson, P. B.; Sweeney, C.; Talbot, R.; Townsend-Small, A.; Yacovitch, T. I.; Zimmerle, D. J.; Hamburg, S. P. Reconciling Divergent Estimates of Oil and Gas Methane Emissions. Proc. Natl. Acad. Sci. 2015, 112 (51), 15597–15602. https://doi.org/10.1073/pnas.1522126112.

---

## Author Comment (AC2)

**Author response to reviewer comments**

**Anonymous Referee # 1**

Omara et al compiled previously reported methane measurement data to study methane emissions from major US oil and gas producing basins, and developed a high spatial resolution emission inventory for 2021. I find the methods solid and the manuscript well-written. I only have some minor comments.

We thank Reviewer 1 for these detailed comments and review of our manuscript. We provide point-by-point responses below.

General comments:

More information about spatial allocation from facility-level level to 10 km*10 km resolution is needed (section 2.4). As some basins have more data than others, how much uncertainty will spatial allocation introduce?

- Our methods for spatial allocation of mean total methane emissions requires information on emission estimates per facility and location of the methane emitting facility. Uncertainties on spatial allocation is therefore a combination of uncertainties in our mean total methane emissions per facility (which is discussed separately in Section 2.5 and in the Results and Discussion section) and uncertainties in the spatial accuracy and completeness of the methane emitting infrastructure location information. These latter sources of uncertainties are difficult to quantify based on available information. We also acknowledge that our spatial allocation represents the mean emissions estimates over the year [2021] and are not intended to characterize methane emissions at a specific point in time, as substantial temporal variability in emissions may be expected given the stochastic character of emissions.

- We have included the following sentences to provide additional clarification regarding our spatial allocation methods and related uncertainties [Section 2.4, page 9]:

  > *"Our spatial allocation of estimated total oil and gas methane emissions is dependent, in part, on the completeness and spatial accuracy of oil and gas infrastructure locations for specific regions and oil and gas basins, for which related uncertainties are difficult to quantify based on available information. Our spatial allocation provides the mean methane emissions estimates for the year 2021 aggregated at each 0.1° × 0.1° grid (~10 km × 10 km) and are not intended to characterize methane emissions at a specific point in time, and are not intended to characterize methane emissions at a specific point in time, where substantial short-term variability in emissions may occur in part due to the stochastic character of facility-level methane emissions."*

Additional analysis:

(1) As age of wells is important to methane emissions, would the authors add a plot to show the correlations between age of wells and methane emissions?

- The available measurement data on facility-level well site methane emissions do not generally include information on age of well sites at the time of measurement, and our model does not directly assess the influence of well site age on methane emissions. In general, the potential for fugitive methane emissions to occur at actively producing well site infrastructure is expected across well sites of varying age, a factor which contributes to a general lack of correlation of well site emissions with age (see, for example, Brantley et al., 2014).

- Using proprietary data from Enverus Prism (www.enverus.com), we computed the mean age of well sites within each 0.1º × 0.1º grid (~10 km × 10 km) on which we aggregated mean total methane emissions. We compute the mean age as the average of the age of all actively producing well sites as of 2021-12-31. Figure AR1 below shows that there is essentially no correlation between the mean age and the mean total methane emissions within each grid cell, consistent with the findings from Brantley et al. (20214). Note, however, that we are not directly assessing correlations here because of limited information which precludes explicit treatment of well site age as a variable in our models (i.e., well site age is generally not reported in the measurement-based data).

[Figure]

**Supplementary Fig. 8.** Assessment of mean total methane emissions within each 0.1º × 0.1º grid (~10 km × 10 km) grid cell and correlation with mean well site age.

(2)  It would be good to add a map in Figure 7 to show the uncertainty from EI-ME emissions.

We include in the Supplemental Information additional maps showing our lower and upper bounds on the mean total methane emissions within each grid cell.

[Figure]

**Supplementary Fig. 9.** Estimated spatial distribution of national methane emissions showing the confidence bounds on the mean total methane emissions: **(a)** lower bound estimate representing the 2.5th

95 percentile within each 0.1º × 0.1º grid and **(b)** upper bound representing the 97.5th percentile within each 0.1º × 0.1º grid. The confidence bounds are based on 500 model simulations of each facility's methane emissions as described in the Main Text.

It's good to have a high spatial resolution emission inventory. Have the authors considered improving the temporal resolution of the inventory? If it is not possible, what are the challenges and how to make it possible?

100

- It is possible to produce the EI-ME inventory at finer spatial scales as our model simulates each facility's mean methane emission rates, which can then be spatially allocated to specific grid sizes if facility location is known. A higher-resolution version of the EI-ME inventory is available from the co-authors upon reasonable request for non-commercial, research purposes.

105

Specific comments:

Title: should specify what year is the inventory for.

- We have updated the title to indicate the inventory year:

110    *"Constructing a measurement-based spatially explicit inventory of US oil and gas methane emissions (2021)."*

L21: should specify which year is the inventory for. And clarify what 'mean emission' represents (average of yearly emission, or average of all the uncertainty iterations).

115    - We have revised this sentence in the Abstract as follows:

*"We then integrate these emissions data with comprehensive spatial data on national oil and gas activity to estimate each facility's mean total methane emissions and uncertainties for the year 2021, from which we develop a mean estimate of annual national methane emissions, resolved at*
120    *0.1° × 0.1° spatial scales (~10 km × 10 km)."*

L24: should add one decimal for '14-18' to be consistent with L23 '15.7 Tg'

- We have revised this sentence as follows, reporting our mean total methane emission estimates to
125    2 significant figures in Abstract:

*"From this measurement-based methane emissions inventory (EI-ME), we estimate total US national oil/gas methane emissions of approximately 16 Tg (95% confidence interval of 14-18 Tg) in 2021 which is ~2 times greater than the EPA Greenhouse Gas Inventory."*

130

L42,59: 'methane emissions' to 'methane emission', please check throughout the manuscript

- We use the plural "emissions" throughout the manuscript because the emission of methane arise from a variety of sources within oil and gas operations, for example, including from well sites to
135    gathering and processing facilities to pipelines, each of which may have different root causes for the emissions (e.g., intentional venting, fugitive leakage, equipment malfunction, etc).

L84: how many wells do not have reported production days?

- Roughly 5% of wells in the database did not have reported production days, even as they reported production in the year. Figure AR3 below shows the histogram of reported production days per well, indicating that the vast majority of wells had reported production days.

[Figure]

**Supplementary Fig. 10**. Reported number of production days per actively producing well in 2021. Onshore US wells only. Analysis based on data from Enverus Prism (www.enverus.com).

- We have revised the following sentence in Methods to include the fraction of wells with no reported production days:

*"For each actively producing well, we derive average well-level oil (barrels per day, bpd), gas (1 thousand cubic feet per day, Mcfd), and combined oil and gas (barrels of oil equivalent per day; 1 boed = 6 Mcfd gas) production rates based on the reported number of production days, and assuming 365 calendar days in the year if production days were not reported, which occurred at <5% of producing wells (Supplementary Fig. 10)."*

L104: need a little more information about how the previously published data are searched, such as keywords used for searching on google scholar (or somewhere else).

- Our focus was on previously published peer-reviewed data on facility-level methane emissions measurements for US oil and gas basins. Our search was conducted primarily based on Google Scholar, including key words reflecting the subject (oil and natural gas methane emissions), geography (US oil and gas basins), measurement methods (ground-based, OTM-33A, tracer flux),

and major facility categories (well sites, compressor stations, processing plants, pipelines, crude oil refineries).

We have revised the following sentence in Methods as follows:

- *"We begin by performing a comprehensive data review and assessment of previously published peer-reviewed data on facility-level methane emissions measurements for US oil and gas basins, leveraging Google Scholar search results based on key words that reflect geography of interest (oil and natural gas methane emissions in the US), measurement methods (ground-based facility-level methods, OTM-33A, tracer flux, mobile transects), and major oil and natural gas facility categories (well sites, natural gas gathering and transmission compressor stations, processing facilities, pipelines, crude oil refineries)."*

L166-168: I'm confused with potential bias accounting, can you provide more information? And what about the uncertainty associated?

- One approach we use to evaluate the representativeness of well sites in the measurement data is based in part on the comparison of the distribution of well site gas production rates with the distribution of the natural gas production rates for the national population of sites. Overlaps in the two distributions provide confidence in the estimated results. As Supplementary Figure 3 shows, we find substantial overlap in the distribution for the sampled sites versus the national population, suggesting reasonable representativeness. However, the peak distribution for the sampled sites is greater than that for the national population of sites, suggesting potential oversampling of the higher producing sites in our sample. To account for this potential bias, we develop methane emission distributions based on production-normalized methane loss rates (methane emission normalized by methane production) with well sites stratified into seven different cohorts based on their production rates as described in greater detail in Section 2.3 and in Figure 1a. The uncertainties associated with our estimates is driven by uncertainties in the modeled distributions, which we assess separately for each cohort of sites as part of the model development. We provide further details for the uncertainty assessment in Section 2.5.

We have moved this paragraph to Section 2.3 which describes the facility-level methane emissions model development and provided additional information on the comparison of the distribution of facility-level production rates for the measured sites with the distribution for the national population of well sites:

> *"In addition, the distribution of facility-level natural gas production rates shows reasonable overlap with that for the national population of non-low production facilities, and the broad range in distribution of facility-level production rates across the national population of sites (~90 Mcfd to >50,000 Mcfd) is well represented in the sampled sites (Supplementary Fig. 3c). However, the distribution of production rates for the sampled sites suggests potential bias toward higher-producing sites relative to the national distribution (Supplementary Fig. 3c). We account for any such potential biases by developing emission models based on production-normalized methane loss rate*

205
*distributions (methane emitted relative to methane produced) across seven cohorts of specific gas production rates (further details below).*

*We develop and use probabilistic emission rate distributions based on production-normalized methane loss rates, which shows a wide range <0.01% to >90% (Figure 1a)*
210
*across all basins (Supplementary Fig. 3d), reflecting, in part, the diversity in production characteristics within and across basins. We use production-normalized methane loss rate distributions because (i) the empirical data across a wide diversity of oil and gas production facilities suggests an inverse relationship in which high-producing facilities exhibit comparatively lower methane loss rates, and vice versa (Figure 1a) and (ii) the*
215
*consolidated dataset includes measurements collected in earlier years before 2021. By using the production-normalized methane loss rate distribution models for specific cohorts of facility-level production rates, we do not model any particular site that is active in 2021 as exhibiting the same emission rate size as observed when measurements were taken in the past, as the empirical data and the model constrains facility-level*
220
*methane loss rates to production levels, which will be time-variant. As such, we provide a necessary constraint on our estimates, effectively adjusting modelled facility-level methane emission rates if production rates have substantially changed over time."*

225
L221-225: As EPA inventory underestimates emissions, how does using EPA emission factors impact your results, and how is 50% uncertainty assumed?

- Our use of the EPA emission factors for pipelines makes it possible for us to estimate the emissions for these sources (for which we have spatial activity data) and provide a more complete inventory in the absence of detailed measurement-based data, even as we acknowledge that these
230
  emission factors are likely biased low. We assume a 50% uncertainty on these emission factors to be conservative, as EPA typically does not report uncertainty assessment for specific facility categories, while the reported uncertainty on the total emissions for all sources is generally approximately +/-20%.

235
- We have revised the following sentence as follows:
  *"Given the scarcity of facility-level measurements for gathering and transmission pipelines, we use the emission factors estimated by the US EPA Greenhouse Gas Emission Inventory (EPA, 2022; 285 kg methane/mile/year and 582 kg methane/mile/year, respectively) and assume normal distributions of emission factors with 50% uncertainty.*
240
  *Our use of EPA's GHGI emission factors for oil and gas pipelines makes it possible to provide a more complete spatially explicit inventory of oil and gas methane emissions (inclusive of gathering and transmission pipelines for which we have geospatial activity data), but likely increases uncertainties in our total methane estimates given potential underestimation in the GHGI emission factors."*

245

Figure 4: (1) why is the yellow bar for EDGAR hatched? (2) I suggest moving the year for each study from the bottom of the figure to the top.

We have revised Figure 4 as follows:

[Figure]

**Figure 4.** Comparison of this study's national estimate of total methane emissions from the oil and gas supply chain with previous measurement-based estimates and bottom-up inventories. The first three bars show the oil and gas methane emissions estimated based on facility-level measurements (this study, Alvarez et al. 2018) and production-sector-only methane emissions estimate by Rutherford et al. (2021) using models developed from component-level measurement data. Blue bars show the estimated emissions for the oil and gas production sector, gold bars show the estimated emissions for the midstream and downstream facilities (compressor stations, processing plants, refineries, gathering and transmission pipelines). Error bars show the estimated 95% confidence bounds on the mean total methane emissions estimates. This study's estimate of total national methane emissions include ~0.1 Tg/year of estimated methane emissions for Alaska. The green bars and the red bars show the satellite-derived estimates for contiguous US based on GOSAT and TROPOMI observations, respectively. The last two bars show the "bottom-up" inventories from EPA GHGI and EDGAR v8 for the contiguous US. In all cases, the year for which methane emissions are estimated are shown on the top x-axis.

Figure 5. The colors are similar and difficult to distinguish.

We have revised Figure 5, reducing the number of classes, for ease of readability.

[Figure]

**Figure 5.** Basin-level differences in modeled mean total methane emissions and comparison with the EPA GHGI (Maasakkers et al., 2023), TROPOMI-derived estimates (Shen et al., 2022) and GOSAT-derived estimates (Lu et al., 2023).

L373: Miller et al.,2023 is missing in the reference list.
We have included Miller et al in the reference list.

Figure 7: consider moving the figure legends outside the maps (now they overlap with each other)

We have revised Figure 7 so that the legend is not intersecting the state/country boundaries.

**References**

Brantley, H. L., Thoma, E. D., Squier, W. C., Guven, B. B., Lyon, D. Assessment of Methane Emissions from Oil and Gas Production Pads Using Mobile Measurements. Environ. Sci. Technol., 48 (24), 14508–14515, https://doi.org/10.1021/es503070q, 2014.

EPA: United States Environmental Protection Agency, Inventory of US Greenhouse Gas Emissions and Sinks, https://www.epa.gov/ghgemissions/inventory-us-greenhouse-gas-emissions-and-sinks (last access: 20 December 2023), 2022.

EDGAR (Emissions Database for Global Atmospheric Research) Community GHG Database, a collaboration between the European Commission, Joint Research Centre (JRC), the International Energy Agency (IEA), and comprising IEA-EDGAR CO2, EDGAR CH4, EDGAR N2O, EDGAR F-GASES version 8.0, European Commission, JRC (Datasets), https://edgar.jrc.ec.europa.eu/report_2023, 2023.

Enverus Prism, www.enverus.com, last accessed February 06, 2024

---

## Author Comment (AC3)

**Author response to reviewer comments**

**Anonymous Referee # 2**

Omara et al. constructed a high-resolution inventory for methane emissions from the U.S. oil and gas industry based on reported site-level measurements. The work provides a baseline that incorporates the best information for future evaluation of oil & gas methane emissions in the U.S., thus an important contribution to the field. I appreciate that the statistical method applied in the study is carefully designed with adequate sophistication. I'd recommend publication of the manuscript in ESSD, after the following comments are addressed.

We thank Reviewer #2 for these detailed comments and review of our manuscript. We provide below point-by-point responses.

1. The title indicates the inventory is for "US oil and gas methane emissions". However, the work is actually for "contiguous US onshore up- and mid-stream oil and gas emissions". The language can be more precise in places like abstract, conclusion, and Section 3.1 (when national totals are compared). While the focus on "onshore up- and mid-stream" is explained in the main text, I do not find any explicit language about the spatial extent (can only be inferred based on Fig. 7 and 8). As Alaska is an important oil & gas production region, I am concerned about if the comparisons are "apple to apple" in e.g. Section 3.1 when varied "national" totals are compared and discussed.

   Our estimates of oil and gas methane emissions are indeed for the continental United States, including Alaska. The full inventory, including estimated emissions for Alaska, is included in the GeoPackage data file (EI_ME_v1.0.gpkg, https://zenodo.org/records/10909191). For Alaska, we estimate total onshore oil and gas methane emissions of ~0.1 Tg in 2021, representing ~0.6% of the estimated national total. As such, including or excluding the estimated onshore-only methane emissions for Alaska does not alter the overall conclusions from our study in comparison with previous studies on national oil and gas methane emissions. We note that Alaska is generally excluded in recent works on satellite-based inversion studies of methane emissions, where primary focus has been on assessing the emissions from the contiguous (lower 48 states) United States (e.g., Shen et al., 2022; Lu et al., 2023; Nesser et al., 2023). We do provide a netcdf for only the contiguous US so as to allow for more of an "apple-to-apple" comparison with these studies.

   We have made revisions throughout the manuscript to clarify, where needed, that our estimates include estimates of ~0.1 Tg/year of for Alaska. For example, in the figure caption for Figure 4, we include the following sentence for clarity:

   *"This study's estimate of total national methane emissions include ~0.1 Tg/year of estimated methane emissions for Alaska."*

2. The method for low-production wells is not described in the manuscript. Reference to Omara et al. (2022) is provided. However, given the importance of low-production sites found in this work, a brief description of the main idea (e.g., method and data source) of Omara et al. (2022) seems necessary.

We have included the following sentences to briefly describe the methods for the estimation of methane emissions from low production sites:

*"Briefly, we use the reported empirical observations (n = 240; Omara et al., 2022) in a hybrid Monte-Carlo and non-parametric probabilistic model that simultaneously estimates the frequency of below-detection-limit sites, the frequency of high-emitting sites representing the top 5% of emitting facilities based on absolute methane emissions, and the distribution of high-emitter methane emissions, while accounting for the weakly observed positive relationship between emission rates and production rates for the bottom 95% of emitting well sites. We integrate this model with spatially explicit activity data on low-production oil and gas well sites in 2021 (Enverus, 2024) to estimate their total methane emissions."*

In addition, there is inconsistency in the current description of the well-site measurements (Table 1, Line 145-146, and Fig. 1a). Table 1 shows that there are n=1153 samples for low-production and non-low production sites combined. But line 145-146 and the caption of Fig.1a indicate that the figure is for non-low production sites only and includes n=1153 samples.

We have revised the Figure 1 Caption to fix the typo in the number of non-low production well sites:

*"Facility-level methane emissions data (percent methane loss rate) as functions of gas production rates (n = 961 non-low production well sites)."*

Line 142: The fraction of methane in produced natural gas should vary greatly from basin to basin. Is there better information for this parameter? What's the impact of this assumption on the uncertainty?

We do expect variability in the fraction of methane in produced natural gas across various basins, given the differences in geologic characteristics. However, the lack of comprehensive spatial data on methane composition across basins limits our ability to assess the impact of this parameter on our estimates of basin-level and national methane loss rates. Our assumption of an average 80% average methane content across basins is informed by estimates from the EPA Greenhouse Inventory, which reports regional variability of 77.1% in the Rocky Mountains region to 91.9% in the West Coast region, with an overall national average of 82.5%. Using an assumed higher methane content in natural gas leads to lower methane loss rate calculation, and vice versa. If we assume the full range of ~77% to 90% of regional variability in methane content, our computed

average methane loss rate ranges from ~2.3% to 2.7%, which falls within our overall 95% confidence bounds of 2.3 to 2.9%.

We have included to following sentence in Section 3.1 for additional information:

*"In 2021, we estimate a national methane loss rate of 2.6% (95% CI: 2.3 – 2.9%) relative to gross natural gas production, assuming an average of 80% methane content in natural gas."*

In Section 2.4, we provide additional clarification on the computation of methane loss rates.

*"We compute basin-level and national methane loss rates as the ratio of estimated basin-level methane emissions to gross methane production in 2021, based on gross natural gas production data from Enverus Prism (Enverus, 2024) and an assumed average methane content of 80% in natural gas. Our assumption of an average 80% methane content in natural gas is informed by regional estimates of methane composition in natural gas based on the EPA GHGI (EPA, 2022). We acknowledge that uncertainties in methane composition across basins likely increases uncertainties in our overall methane loss rate calculations. Further studies on basin-level methane composition are needed to constrain these uncertainties. This methane intensity metric allows for a direct comparison of estimated methane losses relative to gross methane production across different basins. While our use of gross methane production accounts for emissions from associated gas produced during oil operations, the results are not intended to represent lifecycle emission intensities, which are outside the scope of this work."*

3. Line 182-187: (1) Based on the description, it is unclear whether the distribution of fBDL or only the mean of fBDL is used in the "decrement total mean estimate by fBDL" step. (2) fBDL is defined below in L210 for mid-stream facilities, but the concept first appears here but fBDL is not defined.

We now define $f_{BDL}$ in the first paragraph of Section 2.3:

*"For each cohort, we simulate the frequency of finding a site emitting below the method detection limits (reported as zeros or below the method detection limit) through a random bootstrapping procedure, repeated $10^4$ times, with replacement. From this simulation, we develop a frequency distribution for the sites below the detection limits ($f_{BDL}$), which averaged roughly 20% to 30% for all of the cohorts, with the exception of the last production cohort (>10 Mcfd), where the frequency drops to roughly 10 to 20% (Supplementary Fig. 1)."*

We also clarify that $f_{BDL}$ is used to decrement the mean based on random draws from the modelled distribution:

130 *"As some facilities can have emissions below the method detection limits, we decrement the total estimated emission rate based on a randomly sampled frequency of BDL sites ($f_{BDL}$), randomly drawn from the modelled distributions."*

135 4. Section 2.1 Non-SI units are used throughout the text. It'd better to provide a conversion for SI units.

The units used in the manuscript for oil and gas production are standard units used by oil and gas industry in the US. We have provided the following conversion: 1 ft3 = 0.0283 m3 and 1 bbl
140 crude oil ~ 0.136 tonnes.

*"Briefly, we use the monthly well-level oil and gas production data as reported by Enverus Prism (Enverus, 2023), which aggregates public and proprietary data on monthly well-level production. For each actively producing well, we derive average well-*
145 *level oil (barrels per day, bpd; 1 barrel crude oil ~ 0.136 tonnes), gas (1 thousand cubic feet per day, Mcfd; 1 $ft^3$ = 0.0283 $m^3$), and combined oil and gas (barrels of oil equivalent per day; 1 boed = 6 Mcfd gas) production rates based on the reported number of production days, and assuming 365 calendar days in the year if production days were not reported, which occurred at <5% of producing wells (Supplementary Fig. 10)."*
150

5. Line 151: "as a function of"?

We have revised this sentence to read:

155 *"Facility-level methane emissions data (percent methane loss rate) as a function of gas production rate."*

6. Table 2: Shen et al. (2022) results presented in Fig. 5. can also be shown here.

160 We have now included the percent methane loss rate results from Shen et al. (2022) in Table 2.

7. A recent publication by Sherwin et al. (2024) in Nature reported a large dataset of aerial site measurements over US oil & gas basins. A discussion, if possible, can provide interested readers with useful information. For instance, (1) How does this study compare with Sherwin et al.
165 (2024) at the basin level? (2) What's the implication of this large measurement data to the national inventory compilation?

Sherwin, E.D., Rutherford, J.S., Zhang, Z. et al. US oil and gas system emissions from nearly one million aerial site measurements. Nature 627, 328–334 (2024).
170

Sherwin et al. uses data from different snapshot facility-level aerial surveys conducted over multiple years (2017 to 2021) in combination with component-level simulations of missed

emissions (i.e., below detection limits of aerial methods) to estimate the emission size distribution, i.e., the proportion of sites responsible for the majority of emissions in select regions across the US. The focus of the Sherwin et al. study and the methods used are different from the present study's methods and scope. Specifically, the study focuses on characterizing emission size distributions using aerial remote sensing data as opposed to the present study which is focused on the development of high-resolution spatially-explicit total methane emissions at the basin and national scale. Direct comparison with our study's results is limited by these and other caveats noted in the study. We acknowledge that more measurements and analyses are needed, specifically, direct quantification of total area methane emissions in combination with the assessment of the emissions from high emitting facilities will help constrain uncertainties in the assessment of emission size distributions. Interested readers are referred to Williams et al. (2024) for a detailed discussion of emission size distributions and uncertainties for the US upstream and midstream facilities and comparison with different studies that have explored this subject in recent years.

In Section 7 (Conclusions) we emphasize that further improvements to measurement-based methane emission inventories are possible:

[revised manuscript text omitted]